# Neural dynamics underlying self-control in the primate subthalamic nucleus

**Benjamin Pasquereau[1,2]\*, Robert S Turner[3]**

[1]Institut des Sciences Cognitives Marc Jeannerod, UMR 5229, Centre National de la Recherche Scientifique, 69675 Bron Cedex, Bron, France; [2]Université Claude Bernard Lyon 1, 69100 Villeurbanne, Villeurbanne, France; [3]Department of Neurobiology, Center for Neuroscience and The Center for the Neural Basis of Cognition, University of Pittsburgh, Pittsburgh, United States

**Abstract** The subthalamic nucleus (STN) is hypothesized to play a central role in neural processes that regulate self-control. Still uncertain, however, is how that brain structure participates in the dynamically evolving estimation of value that underlies the ability to delay gratification and wait patiently for a gain. To address that gap in knowledge, we studied the spiking activity of neurons in the STN of monkeys during a task in which animals were required to remain motionless for varying periods of time in order to obtain food reward. At the single-neuron and population levels, we found a cost–benefit integration between the desirability of the expected reward and the imposed delay to reward delivery, with STN signals that dynamically combined both attributes of the reward to form a single integrated estimate of value. This neural encoding of subjective value evolved dynamically across the waiting period that intervened after instruction cue. Moreover, this encoding was distributed inhomogeneously along the antero-posterior axis of the STN such that the most dorso-posterior-placed neurons represented the temporal discounted value most strongly. These findings highlight the selective involvement of the dorso-posterior STN in the representation of temporally discounted rewards. The combination of rewards and time delays into an integrated representation is essential for self-control, the promotion of goal pursuit, and the willingness to bear the costs of time delays.

**\*For correspondence:**
benjamin.pasquereau@cnrs.fr

**Competing interest:** The authors declare that no competing interests exist.

## Editor's evaluation

This study provides valuable information regarding the neurophysiological basis of self-control. The authors recorded the single neuron activity in the subthalamic nucleus in Monkeys. The authors found neurons whose activity was modulated by reward magnitudes and delays.

## Introduction

Imagine you are standing in a queue in front of a bakery. How long are you willing to wait for your favorite pastry? Many of us lose patience after about 5 min, while others persevere and keep waiting during longer delays. Our individual ability to delay gratification and maintain self-control depends on an internal process that estimates continuously the trade-off between the desirability of the benefit expected and the cost of waiting (*Ainslie, 1975*). All animals, including humans, prefer to receive rewards sooner rather than later, a phenomenon known as temporal discounting (*Frederick et al., 2002*; *Loewenstein and Prelec, 1993*; *Mazur, 2001*; *Vanderveldt et al., 2016*). Accordingly, people with low discount rates tend to pursue their long-term goals patiently, whereas people with high discount rates often abandon their goals impulsively and move on (*Janakiraman et al., 2011*). In economic behavior, the net payoff for such a cost–benefit dilemma is typically evaluated

by integrating the magnitude of the future reward with a hyperbolic discounting function (*Green and Myerson, 2004*; *Kirby, 1997*; *Loewenstein et al., 1992*). Most studies of the neural correlates of temporal discounting have focused on the task instruction period, the point in a trial when the subject is informed of the size of the reward to be delivered and of the delay in time until its delivery (*Berns et al., 2007*). Far less is known about neuronal activity during the subsequent post-instruction delay period, during which subjects may exhibit varying degrees of patience (e.g., self-control) and anticipation of reward. It is quite possible that neuronal activity related to those factors also evolves dynamically across this time period. For example, as the subjective value of a future reward is updated with the passage of time, the motivation to achieve a delayed goal may vary gradually. Indeed, functional imaging studies in humans suggest that neural activity related to temporal discounting evolves dynamically during a post-instruction delay period in patterns that differ distinctly between brain regions (*Jimura et al., 2013*; *McGuire and Kable, 2015*; *Tanaka et al., 2020*). How such dynamically evolving encodings of temporally discounted subjective value are instantiated at the single-unit level remains poorly understood.

Because the subthalamic nucleus (STN) is thought to be crucial in inhibitory control by preventing impulsivity (*Aron et al., 2016*; *Bonnevie and Zaghloul, 2019*; *Jahanshahi et al., 2015*) and modulating the performance of reward-seeking actions (*Baunez et al., 2007*; *Baunez and Robbins, 1997*), we hypothesized that this structure could contribute to the maintenance of adaptive behaviors by dynamically computing the temporally discounted value. The STN occupies a unique position for translating motivational drives into behavioral perseverance, standing at the crossroads between the basal ganglia indirect pathway and many hyperdirect inputs from prefrontal areas involved in motivational, cognitive and motor functions (*Haynes and Haber, 2013*; *Parent and Hazrati, 1995*). Current functional models of the STN propose that increased activity in the STN extends the time to action initiation by elevating decision thresholds, preventing suboptimal early responses or decisions, especially in situations in which the motivational options are conflicting (*Cavanagh et al., 2011*; *Frank, 2006*; *Mansfield et al., 2011*). In support of these models, a series of lesion studies performed on rats provided causal evidence that STN restrains premature responding in instrumental tasks (*Baunez and Robbins, 1997*; *Wiener et al., 2008*) and controls the willingness to work for food (*Baunez et al., 2005*; *Baunez et al., 2002*). Dysfunctions of STN circuits even produced perseverative actions with a reduced ability to switch between behaviors (*Baker and Ragozzino, 2014*; *Baunez et al., 2007*), making this brain region a good candidate for regulating self-control and delayed gratification. Until now, however, existing evidence is mixed on whether the STN is involved in temporal discounting (*Aiello et al., 2019*; *Evens et al., 2015*; *Seinstra et al., 2016*; *Seymour et al., 2016*; *Uslaner and Robinson, 2006*; *Voon et al., 2017*; *Winstanley et al., 2005*), and no previous study has investigated how STN neurons process value information across delays.

Aside from its role in motor control, clinical studies support the involvement of the STN in motivational functions. In particular, deep brain stimulation (DBS-STN), which is effective at alleviating motor symptoms in parkinsonian patients, may induce a variety of side effects related to altered motivation such as depression, excessive eating behavior, and hypomania (*Berney et al., 2002*; *Castrioto et al., 2014*; *Jahanshahi et al., 2015*; *Voon et al., 2006*). Electrophysiological recordings collected from the STN of these patients have shown low-frequency oscillations (<12 Hz) and spiking activities related to various aspects of reward processing, with neural signals modulated by the magnitude of monetary reward and cost–benefit value attribution (*Fumagalli et al., 2015*; *Justin Rossi et al., 2017*; *Zénon et al., 2016*). In non-human animals, the ability of the STN to represent the subjective desirability of actions has also been evidenced by studies that show neurons firing as a function of the expected reward and the associated effort cost (*Breysse et al., 2015*; *Espinosa-Parrilla et al., 2013*; *Nougaret et al., 2022*). Although substantial effort has been directed to elucidate the role of the STN in valuation-related processes at the time of decision-making (*Cavanagh et al., 2011*; *Coulthard et al., 2012*; *Frank et al., 2007*) and in movement incentive (*Nougaret et al., 2022*; *Tan et al., 2015*; *Zénon et al., 2016*), much less attention has been paid to STN involvement in the computation of temporally discounted value during a waiting period, when behavioral inhibition must be sustained patiently over time. In addition, it is still unclear how these roles for STN in cost–benefit valuation and motivational processing relate to the known organization of this nucleus into anatomically and functionally distinct territories (*Alexander et al., 1990*; *Nambu et al., 2002*; *Parent and Hazrati, 1995*).

To determine whether the STN conveys signals consistent with its predicted role in pursuing delayed gratification, we trained two monkeys to perform a delayed reward task in which animals were required to remain motionless during post-instruction delay periods of varying durations in order to obtain food reward. We hypothesized that STN neurons exhibit a dynamic encoding of temporally discounted value over the time course of the delay period consistent with a continuously evolving value attribution essential for self-control. Here we tested this hypothesis by studying spiking activity in the STN while monkeys performed the task. At the single-neuron and population levels, our results support a role for the STN in temporal discounting and indicate that neural signals underlying the valuation of reward size and delay are integrated dynamically into subjective value along an antero-posterior axis in this nucleus. Such dynamic value integration through the STN may regulate the expression of persistent behaviors for which a continuously evolving cost–benefit estimation is required to monitor and sustain goal achievement.

## Results

### Monkeys' behavior reflects reward size and delay in an integrated manner

Two monkeys (H and C) were trained to perform a delayed reward task in which they were required to align a cursor on a visual target and to maintain this arm posture for varying periods of time before delivery of food rewards (*Figure 1A*). At the beginning of each trial, an instruction cue appeared transiently signaling one of six possible reward contingencies. Cue colors indicated the size of reward (one, two, or three drops of food) and symbols indicated the delay-to-reward (short delay [3.5–5.6 s] or long delay [5.2–7.3 s]). Animals were given the option to reject a proposed trial by moving the cursor outside of the target (e.g., if they did not think it was worth waiting for the expected quantity of reward). In this task, the rejection rate (i.e., the proportion of trials with a failure to keep the cursor in the target) reflects the monkey's motivation to stay engaged in the task and to successfully complete the trial according to its prediction about the forthcoming reward. The six instruction cues effectively communicated six different levels of motivation or subjective value as evidenced by consistent effects on the animals' task performance (*Figure 1B and C*). Rejection rates were affected by both reward size (two-way ANOVAs; monkey H: $F_{(2,666)}$ = 10.47, p<0.001; monkey C: $F_{(2,708)}$ = 5.36, p=0.0049) and delay to reward (H: $F_{(1,666)}$ = 22.62, p<0.001; C: $F_{(1,708)}$ = 8.03, p=0.0047). Although the total proportion of rejected trials differed across monkeys (two-sample *t*-test; $t_{(229)}$ = 3.96, p<0.001), a similar behavioral pattern was observed in both animals during the task. The proportion of rejected trials was higher for smaller rewards and longer delays, while both animals waited more patiently to obtain larger rewards.

Interactive effects between reward size and delay (H: $F_{(2,666)}$ = 19.31, p<0.001; C: $F_{(2,708)}$ = 10.31, p<0.001) revealed an integration of both task parameters to estimate the overall desirability or subjective value of each cost–benefit condition. To characterize how subjective value declined with delay, we fitted the averaged rejection rates to a hyperbolic discounting model (as expressed by *Equation 2*). To be more specific, we inferred the temporal discount factor (*k*) that maximized the inverse relation between each monkey's behavior and the subjective value calculated from a hyperbolic function. Consistent with other monkey studies (*Hori et al., 2021*; *Minamimoto et al., 2009*), the animals' task performance was well approximated by an inverse relation with hyperbolic delay discounting (H: $R^2$ = 0.98; C: $R^2$ = 0.86; *Figure 1D and E*). The resulting discount rates calculated for the two animals were relatively similar in value (*Figure 1F and G*). In comparison, however, monkey C was a bit more impatient with a steeper delay discounting (*k* = 1.62 $s^{-1}$), while the subjective value estimated by monkey H was slightly less impacted by the cost of waiting (*k* = 1.28 $s^{-1}$).

### Controls

We recorded EMGs from different muscles (trapezius, deltoid, pectoralis, triceps, biceps) while monkey H performed the behavioral task. During the post-instruction waiting interval, when the animal remained static, the maintenance of the arm posture resulted in a slight increase in the tonic activity of shoulder muscles (*Figure 1H and I*). As evidenced by a series of two-way ANOVAs (reward × delay, p<0.05/173-time bins), muscle patterns were not altered by reward contingencies. This suggests that monkeys controlled their posture with a constant motor output across trial conditions, independent of reward size and delay. Alternatively, as monkeys were not required to control their

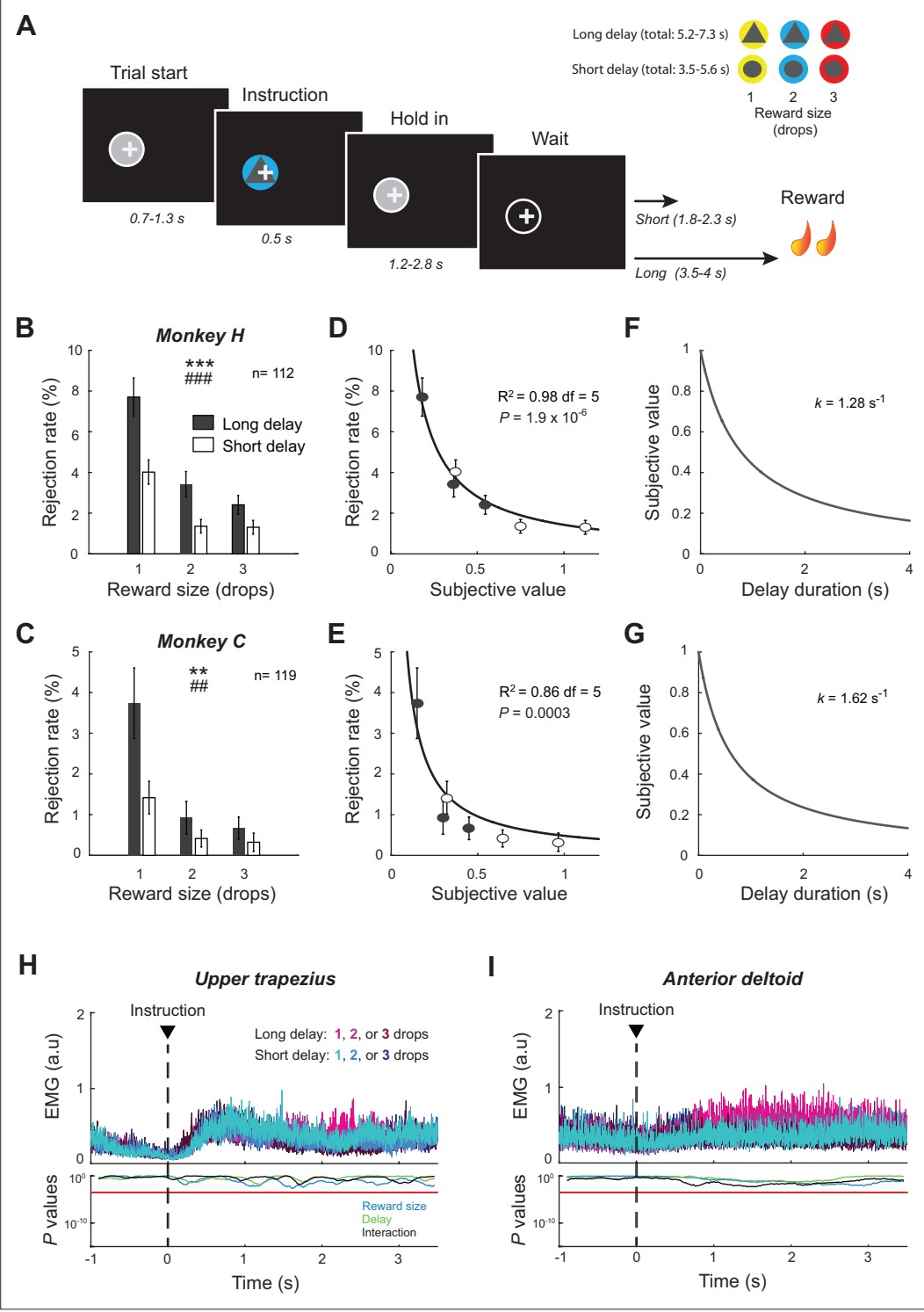

**Figure 1.** Delayed reward task and behavioral performance. (**A**) Temporal sequence of task events. After the monkey initiated a trial by positioning a cursor (+) within a visual target (gray circle), an instruction cue was presented briefly signaling the reward size (one, two, or three drops of food) and the delay-to-reward (short or long). The animal was required to maintain the cursor position over the waiting period to successfully obtain reward. (**B, C**) Rejection rates (mean ± SEM) were calculated and averaged for the six possible reward contingencies across sessions. Measures were affected by both reward size and delay (two-way ANOVA). Size: ***p<0.001, **p<0.01; ddelay: ### p<0.001, ## p<0.01. (**D–G**) For each animal, a temporal discount factor (k) was found that yielded the best fit between averaged rejections rates and the hyperbolic model expressed by

*Figure 1 continued on next page*

*Figure 1 continued*

***Equation 2***. Goodness of fit was evaluated by the coefficient of determination ($R^2$). (**H, I**) EMG signals collected in monkey H were aligned on the presentation of cues. The effects of reward size and delay were examined using a series of two-way ANOVAs. Red lines indicate the statistical threshold ($p < 0.05/173$; Bonferroni correction).

The online version of this article includes the following figure supplement(s) for figure 1:

**Figure supplement 1.** Eye positions varied according to the levels of reward and delay.

gaze while performing the task, we found that their eye positions varied according to the type of trial (***Figure 1—figure supplement 1***). Eye position was affected by both the expected reward size and delay after the presentation of instruction cues (two-way ANOVAs, $p < 0.05/173$-time bins). Reward-by-delay interactions detected in eye position after instruction offset reinforce the view that cost–benefit parameters were integrated into a common valuation by monkeys.

To confirm the ability of our animals to recognize and evaluate appropriately the different instruction cues, the animals also performed a variant of the task that required decision-making. In this variant, the monkey was allowed to choose freely between two alternate reward size or delay conditions. We observed appropriately strong preferences for the cues that predicted large rewards and short delays. Monkeys selected the more advantageous option in terms of reward when the delays were equal (H: 97%, $t_{(14)} = 26.99$, $p < 0.001$; C: 99%, $t_{(16)} = 30.55$, $p < 0.001$) and the more advantageous delay option when reward sizes were held constant (H: 99%, $t_{(11)} = 29.71$, $p < 0.001$; C: 95%, $t_{(10)} = 24.82$, $p < 0.001$).

## Neuronal activity of STN reflects reward size and delay

While the monkeys performed the delayed reward task, we recorded single-unit activity from 231 neurons in the right STN (112 from monkey H; 119 from monkey C). Similar to our previous study (***Pasquereau and Turner, 2017***), STN neurons were identified based on location and standard electro-physiological criteria (***Figure 2A–C***). Most STN neurons exhibited changes in firing rate at one or more times in the task. Approximately 42% of neurons demonstrated a peak in activity in the first second following presentation of the instruction cues, while 32% of neurons exhibited highest discharge rates later during the waiting period (***Figure 2D***). Despite the fact that phasic changes evoked by instruction cues dominated the population-averaged activity (***Figure 2E***), we found that the variability of neuronal activities across the six reward/delay conditions was maintained at an elevated constant level across several seconds of the trial, as evidenced by the Fano Factor shown in ***Figure 2F***. This suggests that task-relevant information was processed by STN neurons not only immediately after the presentation of the instruction cue, but also later in the course of the post-instruction delay period, when the animal was maintaining a stable arm position in anticipation of reward delivery.

To determine whether and when STN neurons were involved in the evaluation of different task conditions, we tested the neural activities for effects of reward size and delay using two-way ANOVAs combined with a sliding window procedure ($p < 0.05/174$-time bins). Because the variability of neuronal activities across task conditions was sustained over time, we analyzed the spiking activity of each neuron in a time-resolved way across a continuous 3.5 s period following the presentation of the instruction cue. Of the 231 neurons recorded, 112 (48%; 21 from monkey H and 91 from monkey C) and 91 (39%; 18 from monkey H and 73 from monkey C) were modulated by reward size and delay, respectively (***Figure 2G–J***). Interestingly, the two types of encoding occurred preferentially during different periods of the trial. Immediately following cue presentation, neurons were strongly influenced by the reward size signaled by the instruction while, later in the trial, as animals endured the waiting period, encoding of delay became more common. Among the 79 neurons (34%; 14 from monkey H and 65 from monkey C) sensitive to both parameters at some point over the course of the trial, 50 (22%; 8 from monkey H and 42 from monkey C) were influenced simultaneously by reward size and delay within the same time bins, thereby reflecting a direct integration of cost–benefit conditions by individual neurons (***Figure 2K and L***). Reward-by-delay interactions were scattered in a roughly uniform distribution across the course of the post-instruction period. Overall, these results suggest that the way STN neurons represented task conditions evolved dynamically across the course of a trial.

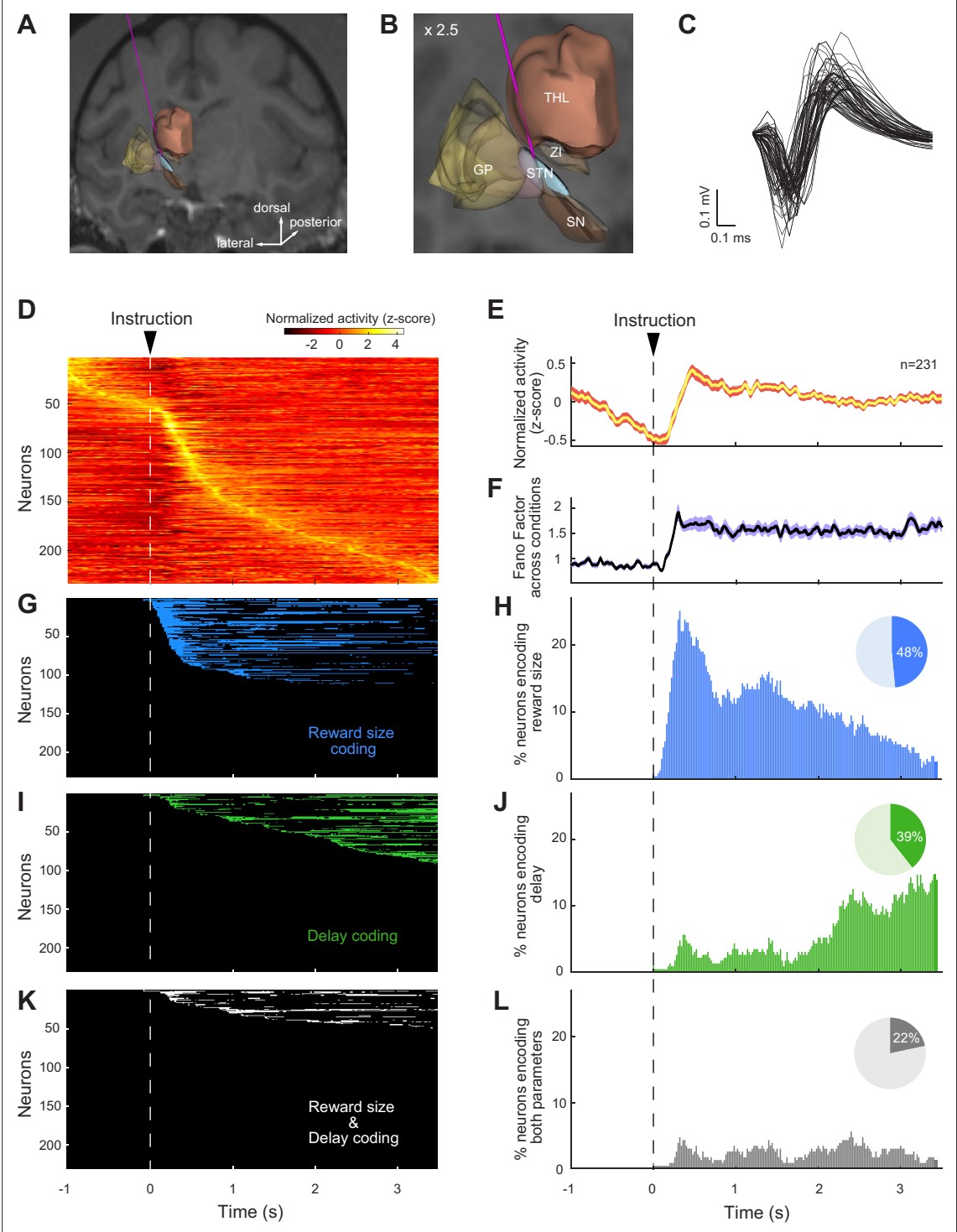

**Figure 2.** Subthalamic nucleus (STN) neurons were modulated by reward size and delay. (**A, B**) Reconstruction of a trajectory used for STN recordings with a structural MRI and high-resolution 3-D templates of individual nuclei derived from *Martin and Bowden, 1996*. Globus pallidus (GP), substantia nigra (SN), zona incerta (ZI), and thalamus (THL). (**C**) Sample of action potential waveforms emitted by STN neurons. (**D**) Color map histograms of neuronal activities recorded from the STN. Each horizontal line indicates neural activity aligned to instruction cues averaged across trial types. Neuronal firing rates were Z-score normalized. (**E**) Population-averaged activities of STN neurons, and (**F**) Fano factors that showed the variability of the neural population ensemble across the six possible reward contingencies. The width of the curves indicates the population SEM. (**G–L**) Influence of reward size and delay on individual neural activities was detected by a series of two-way ANOVAs (p<0.05/173, Bonferroni correction). The time course of encoding of task-relevant information (left column) and the fractions of neurons modulated by reward size and/or delay (right column) were represented for each time bin. Pie charts show the total fraction of STN neurons influenced by reward size (blue), delay (green), or both task parameters simultaneously (gray).

## Dynamic encodings of reward size and delay

To examine how reward size and delay were encoded by individual STN units and how that encoding changed across time in a trial, we performed time-resolved linear regressions with single-unit neural activity as the dependent variable. For each task-related neuron (i.e., neurons encoding at least one task parameter for a least one-time bin, n = 124), we tested whether the firing rate was modulated by the expected reward quantity and the delay to reward delivery (as expressed by *Equation 3*). Because the STN contains an oculomotor territory (*Matsumura et al., 1992*), we included measures of eye movements (i.e., gaze position and gaze velocity) in our model as nuisance variables. (Exclusion of eye parameters from this analysis produced very similar results – *Figure 4—figure supplement 1*.) As illustrated in *Figure 3* with three example neurons, task parameters were encoded in the STN at different stages of the trial following different modalities. (Thresholds for significant regression coefficients were calculated relative to their values during the pre-instruction period using a one-sample *t*-test, df = 46, p<0.05.) Based on the polarity of the regression coefficients $\beta_{Reward}$, we found neurons whose activity transiently indexed reward size by increasing (e.g., neuron #1) or decreasing (e.g., neuron #2) their firing rate. Similarly, by detecting changes in the regression coefficients $\beta_{Delay}$, we found neurons that increased (e.g., neuron #2) or decreased (e.g., neuron #1) their activity as a function of the delay to reward. Neural activities were often influenced in opposite directions by the predicted amount of reward and the delay (positive $\beta_{Reward}$ with negative $\beta_{Delay}$, or vice versa). The specific pattern of task encoding within individual cells, however, often changed over the course of the trial. For example, in the third exemplar unit activity shown in *Figure 3* (right column), the influence of reward size on firing rate (i.e., $\beta_{Reward}$) reversed repeatedly in the post-instruction epoch. This type of variability in the regression coefficients impeded simple approaches for categorization of STN neurons via their pattern of encodings (e.g., positive reward encoding vs. negative).

## Mixed encodings of reward size and delay

To gain deeper insight into how the reward and delay dimensions of the task were integrated by a neuron's activity, we projected the unit's time series of regression coefficients from *Equation 3* ($\beta_{Reward}$ and $\beta_{Delay}$) into a space in which reward size and delay compose two orthogonal dimensions. For each neuron, vector time series were produced in this regression space for significant β values (p<0.05) to capture the moment-by-moment mixture of encodings (*Figure 3C*). In this space, vector angles indicated how a neuron's activity reflected the combined effects of reward size and delay, while vector magnitude captured the strength of the combined encoding. To determine the predominant encoding of these two characteristics (angle and magnitude of moment-by-moment vectors) and their evolution during the post-instruction epoch, we summed across the time-resolved vectors across two consecutive phases of the waiting period (e.g., red dashed lines for phase 1 [0–2 s post-instruction] and phase 2 [2–3.5 s post-instruction] in *Figure 3C*). The angles ($\theta_1$ and $\theta_2$) of the resulting vector sums were used to identify consecutive patterns of activity consistent with, and those inconsistent with, encoding of a temporal discounting of reward value – that is, an encoding in which reward size and delay have opposing effects on firing rate (*Figure 4A*). Over the two phases of the waiting period, some neurons exhibited a consistently positive encoding of reward combined with a negative encoding of delay (vector angles –90° < θ < 0°; referred to as the 'Discounting–' pattern in *Figure 4A*; see, e.g., *Figure 3C*, right), while others modulated their activity in the converse pattern with a negative $\beta_{Reward}$ and positive $\beta_{Delay}$ value (90° < θ < 180°; referred to as 'Discounting+' pattern; see, e.g., *Figure 3C*, middle). Other neurons drastically changed their pattern between phases of the waiting period (*Figure 3C*, left). And some encoded reward size and delay in an additive fashion, inconsistent with a signal reflecting subjective value and referred to here as 'Compounding+' and 'Compounding–' patterns (*Figure 4A*; see, e.g., *Figure 3C*, left). Compounding signals like these are inconsistent with a temporal discounting of reward value and may instead be attributable to extraneous factors such as arousal or attentional engagement.

At the population level, STN neurons exhibited variable mixed signals over the course of the waiting period with, on average, a change in angle of 34° measured between vector sums of phases 1 and 2 (*Figure 4B–D*). First, in phase 1, the neural encoding of reward size and delay parameters predominantly followed a Discounting pattern as evidenced by the fraction of neurons with Discounting– type encodings ($\chi^2$ = 17.1, df = 3, p=0.007; *Figure 4E*), and the longer mean vector magnitude of Discounting– units (one-way ANOVA, $F_{(3,121)}$ = 4.54, p=0.005; *Figure 4G*). Of the 124 task-related

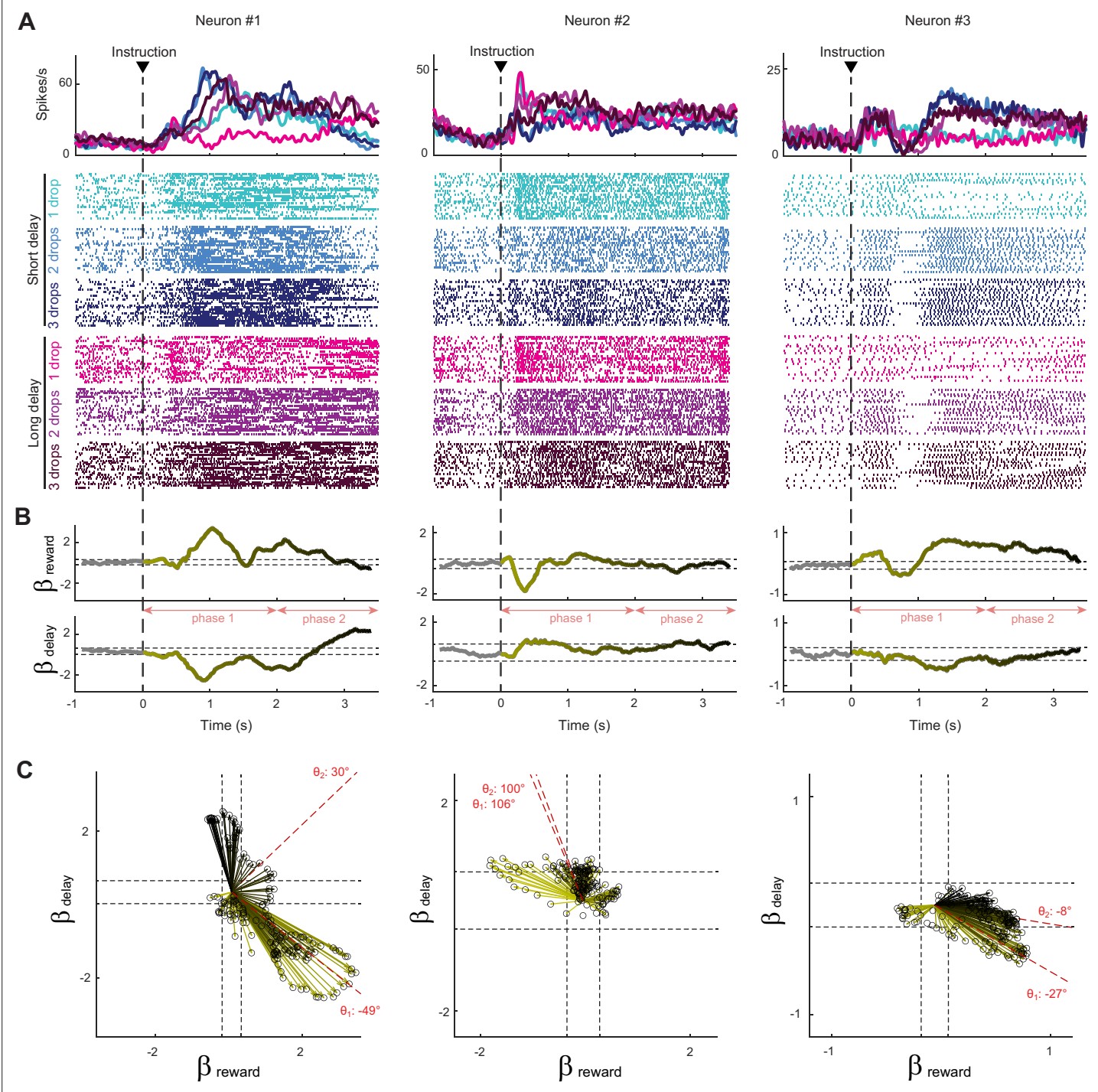

**Figure 3.** Response of subthalamic nucleus (STN) neurons to the six possible reward contingencies. (**A**) The activity of three exemplar neurons that were classified as task-related cells. Spike density functions and raster plots were constructed separately around the presentation of instruction cues for the different cost–benefit conditions. (**B**) A sliding window regression analysis compared firing rates between trial types (as expressed by *Equation 3*). The regression coefficients (yellow-to-black lines) were used to characterize the dynamic encoding of reward size ($\beta_{Reward}$) and delay ($\beta_{Delay}$). The horizontal dashed lines indicate the statistical threshold for significant β values (calculated from the pre-instruction period with a one-sample *t*-test, df = 46, p<0.05). (**C**) Time series of regression coefficients projected into an orthogonal space where reward size and delay composed the two dimensions. Vector time series were produced for significant β values. Black dashed lines indicate statistical thresholds. The angle ( $\theta$ ) of the vector sum (red dashed lines) was calculated to identify how neurons integrated cost–benefit conditions during the two consecutive phases of the waiting period (phase 1: 0–2 s, phase 2: 2–3.5 s).

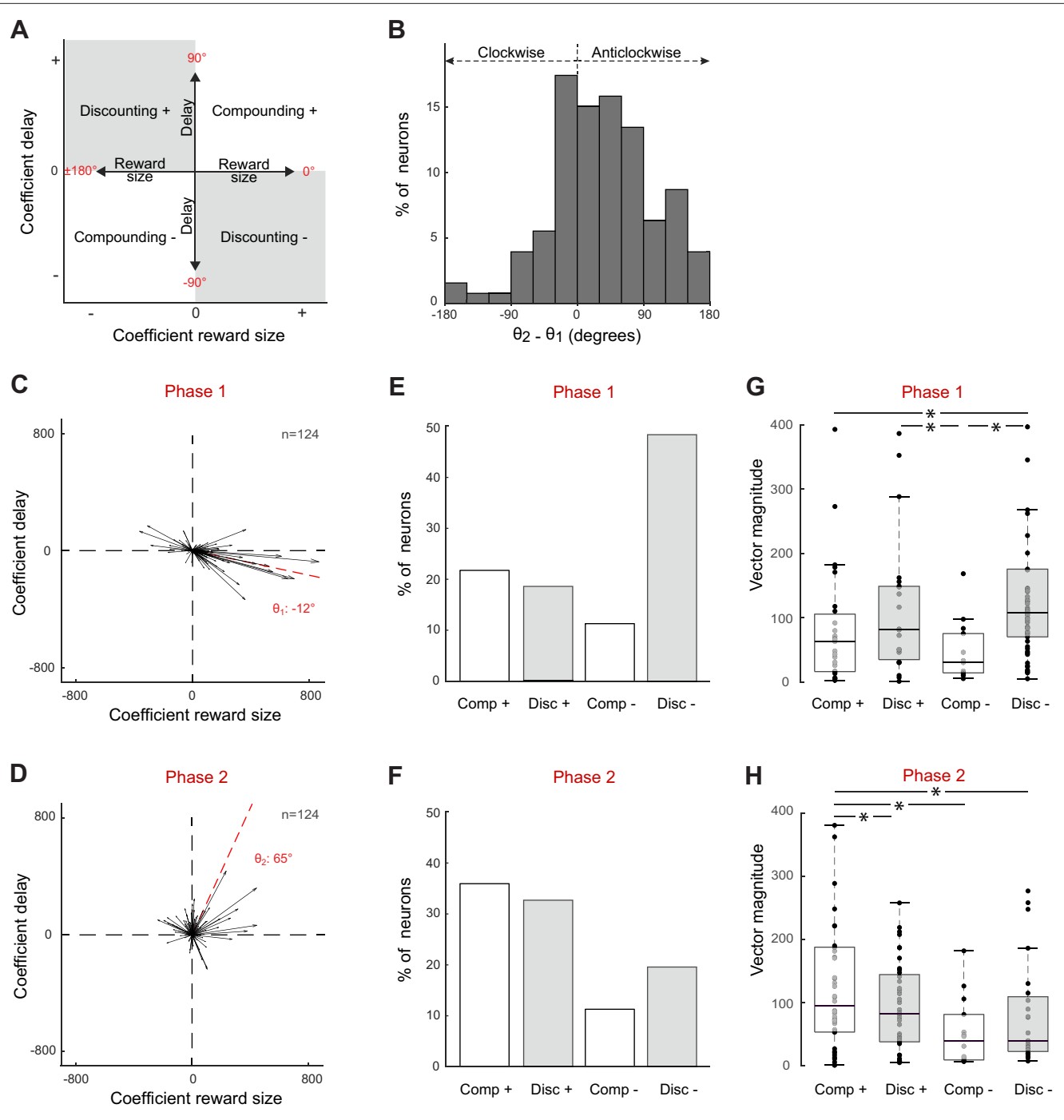

**Figure 4.** Subthalamic nucleus (STN) neurons exhibit mixed signals in phase 1 (0–2 s post-instruction) and phase 2 (2–3.5 s post-instruction). (**A**) Schematic depiction of the regression subspace composed of reward size and delay. Various patterns of neural encoding could be categorized depending on the angle ($\theta$) of vectors: Discounting– (between –90 and 0°); Discounting+ (between 90 and 180°); Compounding+ (between 0 and 90°); Compounding– (between –180 and –90°). (**B**) The angle differences calculated between phases 1 and 2 ($\theta_2 - \theta_1$) show how the neural encodings were modified during the course of the hold period. A positive angle difference corresponds to a turn anticlockwise, while a negative one corresponds to a turn clockwise. (**C, D**) Vectorial encoding of reward size and delay for all task-related neurons in phases 1 (**C**) and 2 (**D**). Vector sums were calibrated by subtracting the mean β values of the pre-instruction epoch and then dividing by 2 SD of this control period. The red dashed lines indicate the population vectors. (**E, F**) Fractions of task-related neurons categorized as Discounting cells (Disc-, Disc+) or Compounding cells (Comp+, Comp-). (**G, H**) Vector magnitudes (mean ± SEM) were compared between different categories of task-related neurons (one-way ANOVA *$F > 2.1$, $p < 0.05$). The central line of the box plots represents the median, the edges of the box show the interquartile range, and the edges of the whiskers show the full extent of the overall distributions.

*Figure 4 continued on next page*

*Figure 4 continued*

The online version of this article includes the following figure supplement(s) for figure 4:

**Figure supplement 1.** Population vectors of the encoding of reward size and delay in phase 1 (**A**) and phase 2 (**B**) using a regression model that excluded eye movement variables ($SC_i = \beta_0 + \beta_R R_i + \beta_D D_i$).

neurons, 60 (48%) increased firing rates as a function of reward size while they decreased according to the temporal delay to reward delivery (i.e., consistent with a Discounting– pattern). The remaining neurons were distributed across the other three encoding patterns. Vector magnitudes for neurons with a Discounting– firing pattern were longer, on average, than those for neurons with either type of Compounding pattern, while the vector magnitude for neurons with a Discounting+ pattern fell in-between. Notably, the angle of the mean vector across all neurons ($\theta_1 = -12°$, *Figure 4C*) showed that, despite the wide diversity of encoding patterns across individual neurons in phase 1, the whole neural ensemble encoded information about reward size and delay in a pattern that was strongly consistent with a temporal discounting of value (i.e., Discounting–). The significance of this bias in the population encoding was supported further by the observation that the angles of individual vectors were distributed in a markedly non-uniform fashion (Rayleigh's test, z = 12.5, p<0.001; *Figure 4C*). Hence, during the first 2 s of the post-instruction period, the neural ensemble combined information related to reward size and delay into a coherent population-scale signal that reflected subjective value according to a Discounting– pattern.

Then, in phase 2, the population encoding of reward size and delay parameters changed drastically to a predominantly Compounding pattern (*Figure 4D*). The fraction of neurons with Compounding+ type encodings increased markedly ($\chi^2$ = 9.9, df = 3, p=0.02; *Figure 4F*), and mean vector magnitude of Compounding+ units was longer than those of units with other types of encoding (one-way ANOVA, $F_{(3,119)}$ = 3.91, p = 0.01; *Figure 4H*). Of the 124 task-related neurons, 42 (34%) increased firing rates during this period as a function of both reward size and delay (i.e., consistent with a Compounding+ pattern), while only 24 (19%) were still categorized as Discounting– type neurons. Inconsistent with a temporal discounting of reward value, the angle of the mean vector across all neurons ($\theta_2 = 65°$, *Figure 4D*) showed that the whole neural ensemble encoded significantly a signal correlated positively with both the reward size and the time delay (Rayleigh's test, z = 40.7, p<0.001). This dynamic shift in the encoding of task-relevant information (i.e., rotation of the vectors between phases 1 and 2) was detected not only at the population level but also in the activities of 27 individual units (22%) in which the encoding pattern switched from Discounting– to Compounding+ in the transition from phase 1 to phase 2, respectively.

Remarkably, the neurons categorized as Discounting– in phase 1 and/or Compounding+ in phase 2 were located preferentially in the most posterior and dorsal portion of the STN (two-sample *t*-test, $t_{(229)}$ > 3.04, p<0.05; *Figure 5*). We did not observe a strict anatomic segregation of neurons with different encoding patterns, but rather an intermixed gradient of cell types along the antero-posterior axis. Discounting– neurons (*Figure 5A–G*) and Compounding+ neurons (*Figure 5H–N*) did not differ from other types of STN neurons with respect to tonic firing rate ($t_{(229)}$ < 1.87, p>0.06) or action potential shape ($t_{(229)}$ < 1.03, p>0.3). Aside from the dorso-posterior bias to their position in the STN, the neurons categorized as Discounting– in phase 1 and/or Compounding+ in phase 2 were indistinguishable from the others.

## Dynamic encodings by the neural population ensemble

We then performed a population-based analysis using all recorded neurons (n = 231) to identify the principal patterns of reward size and delay encoding within the neural ensemble and how they changed dynamically over the course of a trial. After projecting every unit's time series of regression coefficients ($\beta_{Reward}$ and $\beta_{Delay}$) into the orthogonal space composed of the reward size and delay, we used a principal component analysis (PCA) to identify the predominant patterns (i.e., the principal components [PCs]) of encoding across the population. Unlike the single-unit analyses, the estimation of regression coefficients (β values) here was calculated in non-overlapped temporal windows (100 ms) to maintain the independence of β values across time bins. Given that the activity of individual neurons encoded diverse time-varying representations of task-relevant information (see, e.g., *Figure 3*), the PCA identified the patterns that accounted for the greatest variance within the neural population. In the resulting time series of eigenvectors, the angle of the eigenvectors indicated how reward size

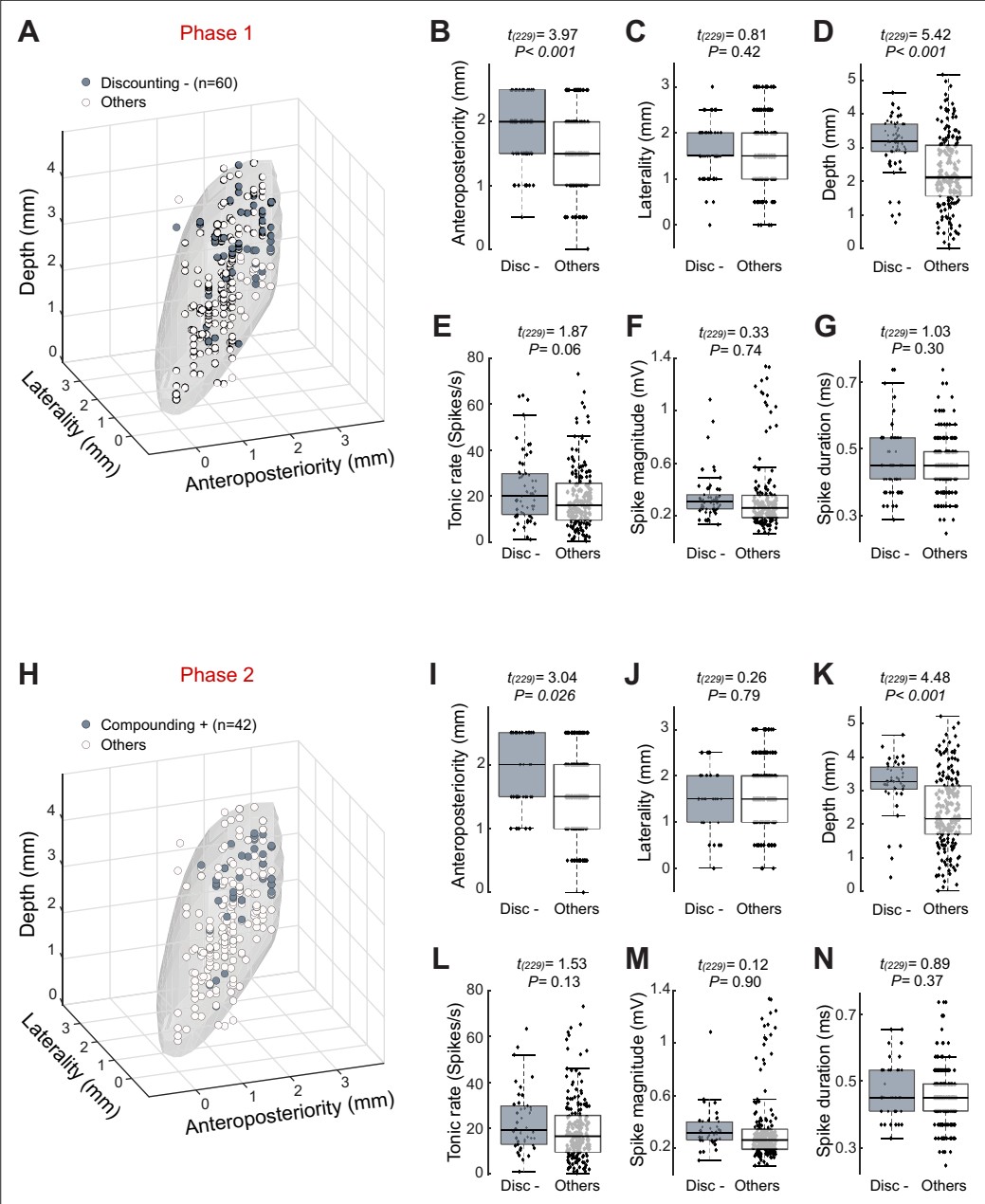

**Figure 5.** Topography of subthalamic nucleus (STN) neurons categorized as Discounting– in phase 1 and Compounding+ in phase 2. (**A**) Three-dimensional plots of cell type distributions based on coordinates from the recording chamber. The template of STN (gray surface) is derived from the atlas (**Martin and Bowden, 1996**). (**B–D**) Comparisons of cell positions show that Discounting– neurons (Disc–) were located more dorsal and posterior than other categories of cell type (two-sample *t*-test). (**E–G**) No differences were found in the tonic rate (**E**), spike magnitude (**F**), and spike duration (**G**) between Discounting– neurons and other cells. The central line of the box plots represents the median, the edges of the box show the interquartile range, and the edges of the whiskers show the full extent of the overall distributions. (**H–N**) Neurons categorized as Compounding+ in phase 2 were located more dorsal and posterior than other cell types.

and delay parameters were integrated at the population level, while the magnitude of eigenvectors captured the relative strength of the encodings in the neural ensemble.

We identified the PCs that accounted for a significant fraction of variance relative to that accounted for by a population of control PCAs (**Figure 6A and B**). Surrogate control PCAs were computed by shuffling neural activity across trials before application of the PCA, thereby scrambling the relationship

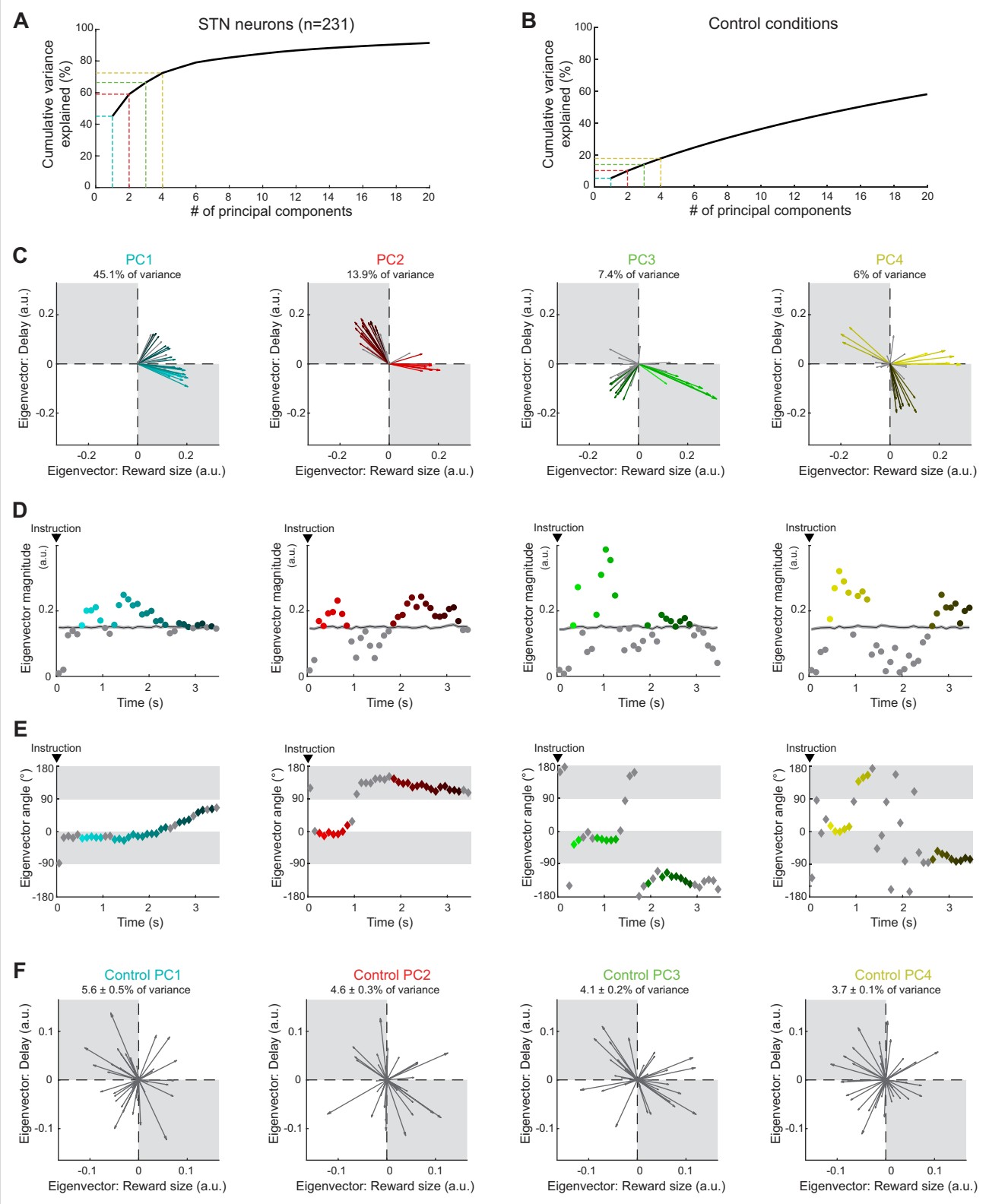

**Figure 6.** Neural population ensemble provides dynamic integration of reward size and delay. (**A**) Cumulative variance explained by principal component analysis (PCA) in the population of all recorded neurons. Dashed lines indicate the percentages of variance explained by the first four principal components (PCs): PC1 (cyan), PC2 (red), PC3 (green), and PC4 (yellow). (**B**) Cumulative variance explained by a control shuffled procedure (data shuffled 1000 times). (**C**) Series of eigenvectors produced for the first four PCs. Eigenvectors capture the moment-by-moment signals in the

*Figure 6 continued on next page*

*Figure 6 continued*

subspace composed of reward size and delay. Colors indicated the eigenvectors with a significant magnitude relative to those calculated from the surrogate control PCAs. (**D, E**) For each PC, eigenvector magnitudes captured the extent of the signals dynamically transmitted (**D**), while angles (−180 to 180°) indicated how the ensemble integrates reward size and delay (**E**). (**F**) Examples of eigenvectors produced by the control shuffled procedure. Percentages of variance explained by the first four PCs were significantly higher than chance (permutation test, p>0.05).

The online version of this article includes the following figure supplement(s) for figure 6:

**Figure supplement 1.** Population-based analysis performed using a sliding window method in the estimation of regression coefficients (200 ms test window stepped in 20 ms).

between task conditions and the neural activity. We found that the first four PCs from the real data exceeded the 95% confidence interval of variances accounted for by PCs from the surrogate control PCAs. In total, the first four PCs explained 72.4% of the total variance (*Figure 6A*). These four PCs were distinct from each other with respect to the pattern represented (*Figure 6E*) and how it evolved across time in a trial (*Figure 6D*). We identified the timing of significant encodings of reward size and delay in PCs by comparing the magnitude of eigenvectors to the 95% confidence interval of those calculated from the surrogate control PCAs (*Figure 6F*). Consistent with our single-unit analyses, the first principal component (PC1, which accounted for 45.1% of variance) corresponded primarily to a temporal discounting of reward value (i.e., a signal that increased with increasing reward size and decreased with increasing delay; $-90° < \theta < 0°$) that later evolved to a Compounding+ pattern (i.e., a signal in which reward size and delay were integrated in an additive manner; $0° < \theta < 90°$). As evidenced by the eigenvector time series of PC1, the neural ensemble encoded a delay-discounted reward value (pointing toward −18°) in a stable manner for approximately 2 s after the presentation of the instruction cue and then rotated progressively toward a nominally significant pattern (pointing toward 44°) in which activity increased with longer delays to reward. Thus, reward size and delay were combined into a common neural ensemble signal that evolved dynamically across the duration of the waiting period. The signal captured by PC2 (second column of *Figure 6*; accounting for 13.9% of variance) differed markedly from that of PC1. In the 1 s after presentation of the instruction cue, the PC2 population signal was primarily sensitive to reward size only, with a transient encoding pointing toward 0°. Then, approximately 2 s after the instruction, PC2 deviated to a robust Discounting+ pattern ($90° < \theta < 180°$) in which reward size and delay had opposing effects on the signal. Consistent with an encoding of temporally discounted values, this Discounting+ pattern was maintained throughout the later half of the waiting period, with a slow progressive rotation of the vector toward an increasing influence of delay the longer the delay period lasted (145–110°, *Figure 6E*). Once again, as with PC1, PC2 combined reward size and delay into an ensemble encoding of integrated value that evolved dynamically across the hold period.

Similarly, the eigenvector properties for PC3 (7.4% of total variance explained; third column of *Figure 6*) revealed a population signal that involved two consecutive temporal phases. Between 0.5 and 1.5 s after presentation of the instruction cue, PC3 approximated a transient Discounting− type pattern of value encoding (pointing toward −24°), after which, PC3 deviated to a relatively stable Compounding− pattern ($-90° < \theta < -180°$) in which neural signals correlated negatively with both the size of reward and the time delay. Finally, the signal extracted by PC4 (6% of variance explained) was characterized by multiple salient signals reflecting reward values only (close to 0°) and Discounting+ ($90° < \theta < 180°$) during relatively short periods after instruction. An additional signal in PC4 detected late in the hold period corresponded to a Discounting− pattern ($-90° < \theta < 0°$). Of interest, *Figure 6— figure supplement 1* shows that a PCA using sliding window analysis (200 ms test window stepped in 20 ms) in the estimation of regression coefficients (β values) produced results that were very similar to those from the non-overlapped temporal window analysis (see % of variance explained per PC), but with a higher number of time bins.

## An anatomical gradient through STN

To test whether the predominant patterns of population encoding (i.e., PCs) varied as a function of location in the STN, we correlated the component scores of neurons with their position along variable anatomical axes (45° rotations along x–y–z axes). One anatomical axis correlated significantly with PC1 and PC2, that is, the two major components consistent with a processing of temporally discounted value (*Figure 7*). Because the scores for PC1 (Spearman's rho = 0.35, p<0.001; *Figure 7*) and PC2 (rho

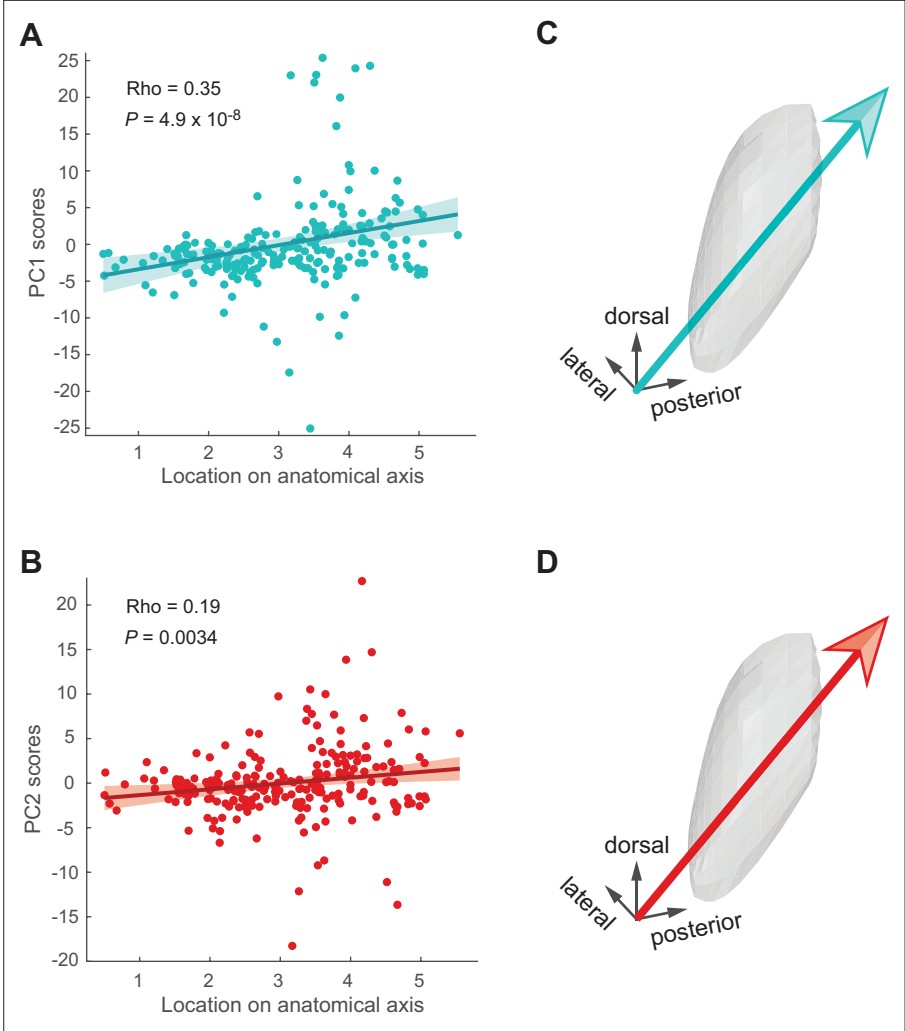

**Figure 7.** Signals vary along the antero-posterior axis of the subthalamic nucleus (STN). (**A, B**) Correlations between component scores (PC1 [cyan] and PC2 [red]) of the neurons and their anatomical position. (**C, D**) Anatomical axis corresponding the most to PC1 (**C**) and PC2 (**D**). Arrows show direction of increasing component scores.

= 0.19, p=0.003) increased along the ventro-anterior to dorso-posterior axis in the STN, this result indicates that reward size and delay were integrated into a common value most strongly in the dorso-posterior STN. No significant correlations were found in the spatial distribution of PC3 and PC4 scores (rho < 0.08, p>0.21), suggesting that these population signals were processed by neurons anatomically intermixed with others in the entire STN.

## Discussion

The present results reveal how STN neurons process temporally discounted value when a behavioral inhibition needs to be patiently sustained over time prior to delivery of a reward. At the single-neuron and population levels, we found a cost–benefit integration between the desirability of the expected reward and the imposed delay to delivery, with signals that dynamically combined both reward-related attributes to form a single integrated value estimate. The computation for such subjective value was increasingly observed along the antero-posterior axis of the STN, revealing that the most dorso-posterior-placed neurons are the most strongly involved in the representation of temporal discounting of value. These results expand our understanding concerning the involvement of STN in motivation and inhibitory control by providing evidence for complex dynamical codings consistent

with a continuous cost–benefit estimation promoting the goal pursuit and the willingness to bear the costs of time delays.

To determine whether the STN conveys dynamic signals consistent with its predicted role in pursuing delayed gratification, we trained monkeys to perform a delayed reward task in which they were required to remain motionless for varying periods of time before the delivery of reward. We designed this task to investigate the ability of our animals to delay gratification and maintain a continuous self-control according to different levels of motivation. Previous studies have already tested monkeys' behavior for delay maintenance (*Evans et al., 2012*; *Evans and Beran, 2007*; *Freeman et al., 2012*; *Freeman et al., 2009*; *Perdue et al., 2015*; *Szalda-Petree et al., 2004*), but none to our knowledge have investigated the underlying neural mechanisms. Our analysis of animals' performance confirmed that different levels of motivation or subjective value influenced their patience and willingness to sustain behavioral inhibition over time (*Figure 1*). As evidenced by the more frequent rejection of trials offering small rewards and long delays, both cost–benefit parameters were integrated by monkeys to complete trials and finally obtain rewards. Consistent with previous findings (*Fujimoto et al., 2019*; *Minamimoto et al., 2009*), the likelihood of staying engaged in the trials for delayed rewards was well approximated by a model that calculates the subjective value with a hyperbolic delay discounting. This shows that monkeys estimated the cost–benefit conditions properly, and that active motivational processes were recruited during the time course of the post-instruction waiting period to maintain their behavioral inhibition.

In both human and non-human animal research, temporal discounting has been traditionally studied via the intertemporal choice task, which pits a small reward available sooner against a large reward available later (*Frederick et al., 2002*). As the delay to the large reward becomes longer, agents tend to start discounting the value of the large reward, biasing their preference toward the small reward available sooner. This choice behavior is considered impulsive and it referred to as a failure of self-control because it would be more economical to wait for the larger reward (*Ainslie, 1975*; *Rachlin, 2004*). To interpret individual choice behavior, the extent to which rewards are devalued over time have been inferred primarily using hyperbolic discounting models (*Mazur, 2001*; *Vanderveldt et al., 2016*). Based on that approach, neuroimaging studies have identified a network of brain areas – involving the striatum, medial prefrontal cortex, and posterior cingulate cortex – activation of which correlates with the subjective value of delayed rewards during decision-making processes (*Kable and Glimcher, 2007*; *McClure et al., 2007*; *McClure et al., 2004*; *Peters and Büchel, 2009*; *Pine et al., 2009*). Consistent with a coding of cost–benefit integration for delays to reward, BOLD activity in these regions increases as the expected amount of a reward increases and, inversely, decreases as a function of the imposed delay to reward. At the single-neuron level, however, electrophysiological results have been more mixed concerning the integration of both reward-related attributes into a common currency value attribution (*Roesch and Bryden, 2011*). While some studies have found strict dissociable representations of both neural signals by different populations of neurons (*Roesch et al., 2007*; *Roesch et al., 2006*; *Roesch and Olson, 2005*), two monkey studies have reported single-unit activities co-modulated by both reward size and delay (*Cai et al., 2011*; *Kim et al., 2008*). Our results confirm and extend to the STN those findings by showing that the cost and benefit dimensions of the task are represented in different ways in different sub-groups of neurons, either independently or in an integrated manner during the waiting period. Respectively, 14 and 5% of STN neurons were exclusively modulated by the expectation of reward size and delay without any co-modulation by the other parameter, maintaining distinct dynamic encodings split between these subsets of cells. Signals related to reward size were represented preferentially shortly after cue presentation, while signals related to delay cost occurred primarily later as animals endured the waiting period (*Figure 2*). The fact we were able to dissociate in the STN these two types of neurons suggests that temporal discounting originates likely from different neural circuits than those that signal expected reward value. Alternatively, 34% of our STN neurons exhibited a dual coding of size and delay information, engendering a representation of both cost and benefit into a common dimension. Among those cells, the most prevalent activity pattern was characterized by opposing effects of reward magnitude and delay on their firing (*Figure 4*). Specifically, STN neurons that fired more strongly for larger anticipated rewards tended to fire less strongly for longer delays (referred as Discounting–), consistent with the computation of temporally discounted value. Despite a high variability across individual neurons, we found that the whole neural ensemble sampled from the STN encoded and combined both reward

size and delay into a strong population signal that corresponded with a temporal discounting of reward value (*Figure 6*). As evidenced by the eigenvector time series of different PCs, reward size and delay were integrated into a common signal that evolved dynamically while animals remained immobile, waiting for the reward. Such dynamics appear consistent with the hypothesized role for STN in self-control and perseverance.

Unlike most studies of temporal discounting that focus on decision-related neural processes at just one time point in the task, we investigated activity during the post-instruction delay period, during which animals were required to exert sustained commitment to an action with a continuous behavioral inhibition. It is quite likely that distinct components of self-control are measured during those two time periods (*Addessi et al., 2013*). Our approach offers the opportunity to investigate the dynamic processes engaged during the post-instruction period to support an animal's ability to delay gratification. An important avenue for future research will be to determine how STN signals, such as those described here, change when animals run out of patience and finally decide to stop waiting. To do this, however, smaller reward sizes and longer delays might be used to promote more escape behaviors during the delay interval. Rejection rates were relatively low in the present version of our task. Given current models where STN is thought to prevent hasty decisions by elevating its activity to pause cortical commands via pallido-thalamocortical circuits (*Cavanagh et al., 2011*; *Frank, 2006*; *Mansfield et al., 2011*), signals underlying a steeper temporal discounting may be observed shortly before behavioral disinhibition if this nucleus effectively drives this type of inhibitory function. In addition, further studies relying on simultaneous multisite recordings are still necessary to clarify the neural origins of the information transmitted by the signals identified here. Although our results show that STN neurons process a dual dynamic coding of size and delay information, it remains undetermined whether the integration between these two reward-related attributes occurs primarily within this nucleus or upstream in other brain areas such as the dorsolateral prefrontal cortex (*Kim et al., 2008*) or the anterior striatum (*Cai et al., 2011*), for instance.

A growing number of animal studies have examined the involvement of STN in motivational functions using single-unit recordings. Neurons with phasic responses evoked by reward-predictive cues and the reward itself have been reported in different species and tasks (*Breysse et al., 2015*; *Darbaky et al., 2005*; *Espinosa-Parrilla et al., 2013*; *Lardeux et al., 2013*; *Lardeux et al., 2009*; *Matsumura et al., 1992*; *Nougaret et al., 2022*; *Teagarden and Rebec, 2007*). However, it has been unclear how this reward processing relates to the known organization of STN into anatomically and functionally distinct territories (*Alexander et al., 1990*; *Nambu et al., 2002*; *Parent and Hazrati, 1995*). The STN receives topographically organized inputs from most regions of the frontal cortex (*Haynes and Haber, 2013*; *Nambu et al., 2002*), the pallidum (*Karachi et al., 2005*; *Shink et al., 1996*), and the parafascicular nucleus of the thalamus (*Sadikot et al., 1992*). Together, tracing studies have indicated that the posterior-dorsal-lateral STN is interconnected with circuits devoted to sensorimotor function, whereas associative- and limbic-related territories are found in progressively more anterior, ventral, and medial regions of the STN (*Haynes and Haber, 2013*; *Mettler and Stern, 1962*; *Monakow et al., 1978*; *Shink et al., 1996*). Based on this, a widely accepted tripartite model divides STN into segregated motor, associative, and limbic regions that are thought to play distinct roles in motor control, cognition, and emotion (*Parent and Hazrati, 1995*). Surprisingly, in previous primate studies, reward-responsive neurons were found to be scattered throughout all parts of the STN (*Darbaky et al., 2005*; *Espinosa-Parrilla et al., 2013*; *Nougaret et al., 2022*), rather than showing a preferential location in the anterior portion that is held to be the zone with strongest connectivity to limbic structures (*Haynes and Haber, 2013*; *Karachi et al., 2005*). That lack of anatomic specificity has been confirmed in human recordings for which reward-modulated neurons were also identified in sensorimotor regions of the STN (*Justin Rossi et al., 2017*; *Sieger et al., 2015*). To date, these paradoxical observations were attributed to a bias in data collection and the inherent observational bias in data collected as part of DBS implantation surgeries due to the posterior-dorsal-lateral location of the target in those surgeries. By accumulating 231 neurons distributed across all portions of the STN, our study provides a more complete survey of this nucleus for regional variations in the representation of reward-related information. At the single-neuron and population levels (*Figures 5 and 7*), we found that the valuation of delayed rewards was represented preferentially in the most dorso-posterior portion of the STN, thereby challenging predictions from the classical tripartite model of STN functional topography. While earlier studies suggested segregated functional subdivisions, now more recent evidence points

toward overlapping territories (*Alkemade and Forstmann, 2014*; *Emmi et al., 2020*). Consistent with a functional convergence within this structure, our results show that motivational signals were conveyed by neurons located in areas traditionally identified as part of the sensorimotor circuits. Because muscle activities were not altered by reward contingencies during the task (*Figure 1H and I*), we assume that these dynamic reward-related signals were primarily processed by cognitive circuits that monitor and manage goal achievement by translating motivational drives into behavioral perseverance. This role in self-control extends our understanding concerning the involvement of STN in the control of impulsive behaviors (*Aron et al., 2016*; *Jahanshahi et al., 2015*; *Zavala et al., 2015*) and the willingness to work for food (*Baunez et al., 2005*; *Baunez et al., 2002*). By providing evidence for dynamic cost–benefit valuation by STN, our results imply that this structure promotes the pursuit of motivated behavior by computing which costs are acceptable for the reward at stake. Further research is needed to determine whether the neural signals identified here causally drive animals' behavior or rather just participate to reflect or evaluate the current situation.

Impatience for reward and lack of perseverance are major facets of many psychiatric disorders. Numerous clinical studies have shown that these maladaptive behaviors are characterized by a disruption in the ability to weigh appropriately the amount of reward against the cost of delay (*Kirby et al., 1999*; *Madden et al., 1997*; *Mitchell, 1999*; *Reynolds, 2006*; *Vuchinich and Simpson, 1998*). For instance, patients with self-control issues such as in gambling disorder (*Alessi and Petry, 2003*), drug addiction (*Kirby and Petry, 2004*; *Madden et al., 1997*; *Washio et al., 2011*), schizophrenia (*Ahn et al., 2011*; *Heerey et al., 2007*; *MacKillop and Tidey, 2011*), depression (*Dombrovski et al., 2012*; *Pulcu et al., 2014*; *Takahashi et al., 2008*), mania (*Mason et al., 2012*), attention deficit hyperactivity disorder (*Barkley et al., 2001*; *Scheres et al., 2010*; *Tripp and Alsop, 2001*), and anxiety disorder *Rounds et al., 2007* have higher delay discounting rates than normal subjects. Our results merge existing lines of evidence that implicate the STN in motivation and inhibitory control, positioning this structure as a potential hub to regulate aberrant reward processing and the capability to postpone. This view agrees with the beneficial effects of DBS-STN on drug addiction reported in recent animal studies (*Pelloux et al., 2018*; *Rouaud et al., 2010*; *Wade et al., 2017*), and with the battery of psychiatric side effects observed in parkinsonian patients after electrode implantation within the STN (*Castrioto et al., 2014*). However, an open question remains on how the STN contributes in these pathological states to distort defective valuation processes in terms of cost–benefit trade-off.

## Materials and methods

### Animals

Two rhesus monkeys (monkey C, 8 kg, male; and monkey H, 6 kg, female) were used in this study. Procedures were approved by the Institutional Animal Care and Use Committee of the University of Pittsburgh (protocol number: 12111162) and complied with the Public Health Service Policy on the humane care and use of laboratory animals (amended 2002). When animals were not in active use, they were housed in individual primate cages in an air-conditioned room where water was always available. The monkeys' access to food was regulated to increase their motivation to perform the task. Throughout the study, the animals were monitored daily by animal care staff and veterinary technicians for evidence of disease or injury and body weight was documented weekly. If a body weight <90% of baseline was observed, the food regulation was stopped.

### Behavioral task

Monkeys were trained to perform the delayed reward task with the left arm using a torquable exoskeleton (KINARM, BKIN Technologies, Kingston, Ontario, Canada). This device had hinge joints aligned with the monkey's shoulder and elbow and allowed the animal to make arm movements in the horizontal plane. Visual cues and cursor feedback of hand position were presented in the horizontal plane of hand movements by a virtual-reality system. A detailed description of the apparatus can be found in our previous studies (*Pasquereau and Turner, 2015*; *Pasquereau and Turner, 2013*).

In our delayed reward task (*Figure 1A*), the monkey was required to align the cursor on a visual target (radius, 1.8 cm) and to maintain this position for varying periods of time before delivery of the food reward. In total, six combinations of reward size and waiting delay were used. A trial began when a gray-filled target appeared (the same location for all trials) and the animal made the appropriate

joint movements to place the cursor in this circle. Maintenance of the cursor within the target required the animal to actively stabilize the posture of both shoulder and elbow joints in the horizontal plane. While the monkey maintained its arm position, an instruction cue was displayed over the gray-filled target for 0.5 s. After a variable interval (1.2–2.8 s), the gray fill disappeared from the circle, cueing the animal to remain motionless during an additional delay until the reward delivery. For the instruction cues, cue colors indicated the size of reward (one, two, and three drops of food) and symbols indicated the delay duration that the animal would have to wait before reward delivery (short delay [3.5–5.6 s] and long delay [5.2–7.3 s]). The Gaussian distributions of these two delay ranges overlapped for 9% of trials. Cue colors were calibrated to have the same physical brightness (~30 cd/m$^2$). The six unique cue types (3 reward sizes × 2 delay ranges) were presented in pseudo-random order across trials with equal probability. At the end of each successful trial, food reward was delivered via a sipper tube attached to a computer-controlled peristaltic pump (1 drop = ~0.5 ml, puree of fresh fruits and protein biscuits). The trials were separated by 1.5–2.5 s intertrial intervals, during which the screen was black. Failures to maintain the cursor in the start position (radius = 1.8 cm) during the interval between instruction cue and reward delivery were counted as errors. After an error, the rejected trial was aborted and a blank screen appeared (1 s), followed by an intertrial interval.

To confirm the ability of our monkeys to evaluate appropriately the different instruction cues, we also trained them to perform a decision task in which they were allowed to choose freely between two alternate reward size or delay conditions. The overall structure of this task was similar to the delayed reward task except two instruction cues were presented simultaneously. The location of the cues alternated randomly between left and right sides. The animal chose its preferential option by positioning and maintaining the cursor on one of the targets. The pair of cues presented on a single trial differed only along one dimension, presenting either a reward-based decision (different colors but the same symbol) or a delay-based decision (different symbols but the same color).

Before the start of data collection, we trained the monkeys to perform these two behavioral tasks for more than 6 mo. Neuronal data were not collected during performance of the decision task.

## Surgery

After reaching asymptotic task performance, animals were prepared surgically for recording using aseptic surgery under Isoflurane inhalation anesthesia. An MRI-compatible plastic chamber (custom-machined PEEK, 28 × 20 mm) was implanted with stereotaxic guidance over a burr hole allowing access to the STN. The chamber was positioned in the parasagittal plane with an anterior-to-posterior angle of 20°. The chamber was fixed to the skull with titanium screws and dental acrylic. A titanium head holder was embedded in the acrylic to allow fixation of the head during recording sessions. Prophylactic antibiotics and analgesics were administered post-surgically.

## Localization of the recording site

The anatomical location of the STN and proper positioning of the recording chamber to access it were estimated from structural MRI scans (Siemens 3T Allegra Scanner, voxel size of 0.6 mm). An interactive 3D software system (Cicerone) was used to visualize MRI images, define the target location, and predict trajectories for microelectrode penetrations (*Miocinovic et al., 2007*). Electrophysiological mapping was performed with penetrations spaced 1 mm apart. The boundaries of brain structures were identified based on standard criteria including relative location, neuronal spike shape, firing pattern, and responsiveness to behavioral events (e.g., movement, reward). By aligning microelectrode mapping results (electrophysiologically characterized X–Y–Z locations) with structural MRI images and high-resolution 3-D templates of individual nuclei derived from an atlas (*Martin and Bowden, 1996*), we were able to gauge the accuracy of individual microelectrode penetrations and determine chamber coordinates for the STN.

## Recording and data acquisition

During recording sessions, a single glass-coated tungsten microelectrode (impedance: 0.7–1 MOhm measured at 1000 Hz) was advanced into the target nucleus using a hydraulic manipulator (MO-95, Narishige). Neuronal signals were amplified with a gain of 10K, bandpass filtered (0.3–10 kHz), and continuously sampled at 25 kHz (RZ2, Tucker-Davis Technologies, Alachua, FL). Individual spikes were sorted using Plexon off-line sorting software (Plexon Inc, Dallas, TX). The timing of detected spikes

and of relevant task events was sampled digitally at 1 kHz. Horizontal and vertical components of eye position were recorded using an infrared camera system (240 Hz; ETL-200, ISCAN, Woburn, MA). For electromyographic recordings (EMG, in monkey H only), pairs of Teflon-insulated multi-stranded stainless-steel wires were implanted into multiple muscles during 12 training sessions. EMG signals were differentially amplified (gain = 10K), band-pass filtered (200 Hz to 5 kHz), rectified, and then low-pass filtered (100 Hz).

## Analysis of behavioral data

We analyzed the way the animals performed the delayed reward task to test whether the behavior varied according to the levels of reward and delay. As the monkey maintained its arm position for varying periods of time before to obtain rewards, the sum of errors per session was the main variable of interest. Rejection rates were calculated by dividing the number of errors by the total number of trials for each task condition. Two-way ANOVAs were used to test these rejections for interacting effects of reward size (one, two, or three drops) and delay (short or long delay duration). Modulation in rejection rates reflected the monkey's motivation to stay engaged in the task and to fully complete the different types of trials by maintaining the correct arm position. In this study, the motivational or subjective value linked to each task condition was estimated by integrating the forthcoming reward size and the delay discounting. The subjective value of delayed reward is commonly formulated as a hyperbolic discounting model (*Green and Myerson, 2004*; *Mazur, 1987*) as follows:

$$SV = \frac{R}{1+kD} \tag{1}$$

where *SV* is the subjective value (i.e., the temporally discounted value), *R* is the reward size, *k* is a discount factor that reflects an individual animal's sensitivity to the waiting cost, and *D* is the delay to the reward. In previous studies (*Hori et al., 2021*; *Kobayashi and Schultz, 2008*; *Minamimoto et al., 2009*), this hyperbolic discounting model has been shown consistently to fit to monkey behavior better than an exponential function. Because the number of errors in task performance is inversely related to the subjective value (*Fujimoto et al., 2019*; *Minamimoto et al., 2009*), we have inferred the subjective value in each monkey by fitting the average rejection rate to the following model:

$$E = \frac{1+kD}{aR} \tag{2}$$

where *E* is the rejection rate, *R* is the reward size, *D* is the delay, *k* is a discount factor, and *a* is a monkey-specific free parameter. We estimated the best pair of free parameters (*k* and *a*) with the MATLAB function 'fminsearch' that provided the maximum-likelihood fit. Goodness of fit was evaluated by the coefficient of determination ($R^2$). Because of the limited number of task conditions (only two delay ranges), we assumed a linearity in the estimation of reward value.

As monkeys were not required to control their gaze while performing the task, we tested whether eye positions varied according to the levels of reward and delay. For this analysis, we combined horizontal and vertical components of eye position to obtain tangential coordinates, and the potential interaction with task parameters was examined using a two-way ANOVA combined with a sliding window procedure (200 ms test window stepped in 20 ms). The threshold for significance was corrected for multiple comparisons ($p < 0.05$/n-time bins; Bonferroni correction). The same statistical method was used to analyze EMGs, with series of two-way ANOVAs performed to test for effects of reward size and delay. All of the data analyses were performed using custom scripts in the MATLAB environment (The MathWorks, MA).

## Neuronal data analysis

Neuronal recordings were accepted for analysis based on electrode location, recording quality (signal/noise ratio of >3 SD) and duration (>120 trials). The width of action potential waveforms was calculated as the interval from the beginning of the first negative inflection (>2 SD) to the subsequent positive peak, and the magnitude of the biphasic spike waveforms was measured between maxima and minima. Trials with errors were excluded from the analysis of neuronal data. Continuous neuronal activation functions (spike density functions [SDFs]) were generated around instruction cues (−1–3.5 s) by convolving each discriminated action potential with a Gaussian kernel (20 ms variance). Mean peri-event SDFs (averaged across trials) for each of the reward and delay conditions were constructed. A

neuron's baseline firing rate was calculated as the mean of the SDFs across the 1 s epoch preceding cue instruction. The Fano factor, defined as the variance-to-mean ratio of firing rates, was used to measure the variability of neuronal activities across the six trial conditions. For each single-unit activity, we tested for effects of reward size and delay using two-way ANOVAs combined with a sliding window procedure (200 ms test window stepped in 20 ms). Specifically, we extracted single-trial spike counts across a series of 200 ms windows and investigated for each step whether the neuronal activity was influenced by (1) reward size, (2) delay to reward, or (3) both task parameters. The ANOVA identified any interacting effects of reward size and delay. The threshold for significance in those ANOVAs was corrected for multiple comparisons using the Bonferroni correction ($p < 0.05$/n-time bins). Application of the same ANOVA analysis and statistical threshold ($p < 0.05$/n-time bins) to neuronal data during the 1 s period of the pre-instruction control epoch (i.e., from a time period when the animal had no information about the upcoming reward and delay) resulted in 0.12% of type 1 (false-positive) errors, thereby confirming that the threshold for statistical significance was appropriate. A neuron was judged to be task-related if its firing rate reflected a significant encoding of the reward size and/or delay for a least one-time bin.

To test how individual task-related neurons dynamically encoded the forthcoming reward and delay discounting through trials, we used time-resolved multiple linear regressions. We tested whether trial-to-trial neuronal activity was modulated simultaneously by the size of reward ($R$), the delay duration ($D$), and measures of eye movements made in the task (tangential position [$P$] and tangential velocity [$V$]) to control for possible relationships of STN activity with gaze control. For each task-related neuron, we counted spikes ($SC$) trial-by-trial within a 200 ms test window that was stepped in 20 ms increments across the 3.5 s period following onset of the instruction cue. For each bin, we applied the following model:

$$SC_i = \beta_o + \beta_R R_i + \beta_D D_i + \beta_P P_i + \beta_V V_i, \tag{3}$$

where all regressors for the ith trial were normalized to obtain standardized regression coefficients (Z-scored in standard deviation units). The β coefficients for each task-related unit were estimated using the 'glmfit' function in MATLAB. The threshold for significance of individual test β values was determined by comparing them against a population of 46 control β values calculated by applying the same sliding window linear regression approach to spike counts from a 1 s pre-instruction control epoch (one-sample t-test, df = 46, $p < 0.05$).

To characterize how individual neurons integrate task parameters dynamically, time series of regression coefficients of *Equation 3* ($\beta_R$ and $\beta_D$) were projected into an orthogonal space where reward size and delay composed the two dimensions (axes) of interest. In this regression space, we used significant points (pairs of $\beta_R$-$\beta_D$) to produce vector time series originating from the control value of the pre-instruction epoch (averaged across time bins). Vectors were generated with significant regression coefficients calculated during the course of the hold period. In this context, vector angles (−180 to 180°) indicated how the neural activity encodes and combines reward size and delay, while vector magnitudes captured the strength of the signals transmitted. Considering these two characteristics (direction and magnitude of vectors), we summed all time-resolved vectors to identify the predominant encoding of task parameters for each neuron analyzed. Various patterns of neural encoding could be categorized from the angle (θ) of the resultant vector sum. For instance, θ indicated whether the neural activity was correlated (positive coding) or anticorrelated (negative coding) with the reward size, the delay to reward, or both. The angle of these vectors was interpreted as follows: 0° (positive $\beta_R$ values) indicated an exclusive positive reward size coding; ±180° (negative $\beta_R$ values) indicated an exclusive negative reward size coding; 90° (positive $\beta_D$ values) indicated an exclusive positive delay coding; and −90° (negative $\beta_D$ values) indicated an exclusive negative delay coding (*Figure 4A*). In this way, a θ between −90 and 0°, or between 90 and 180°, indicated a coding of both reward size and delay consistent with a temporally discounted value in which benefit and cost parameters have opposing effects on the subjective encoding (i.e, positive $\beta_R$ with negative $\beta_D$, or vice versa). In contrast, a θ between 0 and 90°, or between −180 and −90°, indicated a coding of reward size and delay in which the two parameters are integrated in a compound signal inconsistent with a temporally discounted value (i.e., positive $\beta_R$ and $\beta_D$, or negative $\beta_R$ and $\beta_D$).

For population-based figures (*Figure 4*), vectors were standardized between neurons by subtracting the mean values of the pre-instruction epoch and then dividing by 2 SD of this control period. The

population encoding was considered significant if the distribution of the individual vector angles was non-uniform (Rayleigh's test, p<0.001). The population vector was calculated by vectorially summing standardized cell vectors.

We then performed an alternative population-based analysis to characterize the predominant patterns of neural encoding in the STN using all recorded neurons. We used a PCA to identify patterns of encoding (dimensions) of the neuronal ensemble in the polar orthogonal space composed of the reward size and delay. Our procedure was analogous to the analyses performed by *Yamada et al., 2021*. A two-dimensional data matrix $X$ of size $N_{(neuron)} \times N_{(CxT)}$ was prepared with regression coefficients of *Equation 3* (series of $\beta_R$ and $\beta_D$), in which rows corresponded to the total number of neurons and columns corresponded to the number of conditions ($C$, reward sizes and delays) multiplied by the number of time bins ($T$). Unlike the single-unit analyses, the estimation of regression coefficients ($\beta$ values) here was calculated in non-overlapped temporal windows (100 ms) to maintain the independence of $\beta$ values across time bins. A series of eigenvectors was obtained by applying PCA once to the data matrix $X$. In our analysis, the eigenvectors represent vectors at different time bins in the orthogonal space composed of reward size and delay. PCs were estimated using the 'pca' function in MATLAB. As was the case for the vectors computed for single-units, vector angles (−180 to 180°) indicated how the ensemble encodes and combines reward size and delay, while vector magnitudes captured the strength of the signals transmitted. Adequate performance of PCA was estimated with the percentages of variance explained by PCs. To test whether the PCA performance was significant, we constructed a surrogate control population of PCs in which the neural activity ($SC$) was shuffled across trials before application of *Equation 3*. Consequently, in the surrogate data the linear projection of neural activity into the regression subspace was randomized, eliminating any coherent modulation of activity with task parameters (reward size and delay) in the matrix $X$. We built a surrogate control population of PCs by repeating the shuffling procedure 1000 times and then compared percentages of variance explained by actual PCs against the 95% confidence interval of variances accounted for by the population of surrogate PCs. The percentage of variance accounted for by an actual PC was deemed significant if its value fell outside of the 95% confidence interval of the population of surrogate PCs. In addition, we identified the timing of significant encodings of reward size and delay in PCs by comparing the magnitude of eigenvectors to the 95% confidence interval of those calculated from the population of surrogate PCs.

To test how PCs mapped onto anatomical axes in STN, we correlated the component scores of the neurons with their locations (x–y–z coordinates). We performed this in standard stereotaxic space and also for a rotated set of anatomical axes (45° rotations along x–y–z axes). The rotated axis that best correlated with the scores of a component (maximal Spearman's rho) was identified as the optimal axes for explaining that component's variance.

## Additional information

### Funding

| Funder | Grant reference number | Author |
| --- | --- | --- |
| NIH | NIH R01 NS113817-01 | Robert S Turner |
| NIH | NIH R01 NS091853-01 | Robert S Turner |

The funders had no role in study design, data collection and interpretation, or the decision to submit the work for publication.

### Author contributions

Benjamin Pasquereau, Conceptualization, Formal analysis, Investigation, Methodology, Writing – original draft; Robert S Turner, Conceptualization, Funding acquisition, Validation, Writing - review and editing

### Author ORCIDs

Benjamin Pasquereau http://orcid.org/0000-0003-2855-0672
Robert S Turner http://orcid.org/0000-0002-6074-4365

## Ethics

Two rhesus monkeys (monkey C, 8 kg, male; and monkey H, 6 kg, female) were used in this study. Procedures were approved by the Institutional Animal Care and Use Committee of the University of Pittsburgh (protocol number: 12111162) and complied with the Public Health Service Policy on thehumane care and use of laboratory animals (amended 2002). When animals were not in active use, they were housed in individual primate cages in an air-conditioned room where water was always available. The monkeys' access to food was regulated to increase their motivation to perform the task. Throughout the study, the animals were monitored daily by an animal research technician or veterinary technician for evidence of disease or injury and body weight was documented weekly. If a body weight <90% of baseline was observed, the food regulation was stopped.

## Decision letter and Author response

Decision letter https://doi.org/10.7554/eLife.83971.sa1
Author response https://doi.org/10.7554/eLife.83971.sa2

---

# Additional files

## Supplementary files

• MDAR checklist

## Data availability

Data analysed during this study are available at https://github.com/benjaminpasquereau/Neural-dynamics-underlying-self-control-in-the-primate-subthalamic-nucleus, (copy archived at swh:1:rev:42f954c68a8b2faf38c0353364568d1bd4c403aa).

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
