## [Editor Report]

This study provides valuable information regarding the neurophysiological basis of self-control. The authors recorded the single neuron activity in the subthalamic nucleus in Monkeys. The authors found neurons whose activity was modulated by reward magnitudes and delays.

---

## [Decision Letter]

**Decision letter after peer review:**

Thank you for submitting your article "Neural dynamics underlying self-control in the primate subthalamic nucleus" for consideration by *eLife*. Your article has been reviewed by 3 peer reviewers, and the evaluation has been overseen by a Reviewing Editor and Michael Frank as the Senior Editor. The following individual involved in the review of your submission has agreed to reveal their identity: Hiroshi Yamada (Reviewer #3).

Essential revisions:

The subthalamic nucleus (STN) is thought to play important roles in self-control. In this study, the authors sought to characterize the activity of STN neurons while monkeys performed a task in which they had to withhold their response during a delay period whose length was defined by a specific cue. The reviewers thought that this study contains potentially important results but they raised various concerns in some analyses, which reduced their enthusiasm for the present study. We hope that the authors can address the following essential issues.

1) To identify neurons whose activity was modulated by reward size or delay, the authors analyzed many time bins (e.g. 188 sliding windows). The results must be properly corrected for multiple comparisons, yet the method used for correction is not fully described. It is therefore unclear whether the conclusions are well supported by the data.

2) The authors examined the STN activity aligned to the start of the delay and also aligned to the reward. Most of the "delay encoding" in the STN activity was observed near the end of the waiting period. The trouble with the analysis is that a neuron that responded with exactly the same response on short and long trials could appear to be modulated by delay. This is easiest to see with a diagram, but it should be easy to imagine a neural response that quickly rose at the time of instruction and then decayed slowly over the course of 2 seconds. For long trials, the neuron's activity would have returned to baseline, but for short trials, the activity would still be above baseline. As such, it is not clear how much the STN neurons were truly modulated by delay.

3) Another concern is the presence of eye movement variables in the regressions that determine whether a neuron is reward or delay encoding. If the task variables modulated eye movements (which would not be surprising) and if the STN activity also modulated eye movements, then, even if task variables did not directly modulate STN activity, the regression would indicate that it did. This is commonly known as "collider bias". This is, unfortunately, a common flaw in neuroscience papers.

The reviewers raised a number of additional concerns and suggestions as detailed below. Please also address them as much as possible.

*Reviewer #1 (Recommendations for the authors):*

There are a number of issues I noted:

The claim "Accordingly, people with low discount rates tend to pursue their long-term goals patiently, whereas people with high discount rates often abandon their goals impulsively and move on." needs a citation.

The phrasing of "Until now, however, existing evidence is mixed on whether the STN is causally involved in temporal discounting" implies that this paper may provide causal evidence, which it does not.

The model of subjective value here assumes linearity in reward value; this should be made explicit in the text and supported by any evidence you have. It also assumes that the asymptotic rejection rate is zero, which may bias the estimate of k.

No analysis relates to the transition from the gray-filled circle to the unfilled circle and it is unclear of the relevance of this visual cue to the task design.

It is unclear what 'as a control' means when comparing the fits of the models described in Eq. 3 and Eq. 4 or what the relevance of this statistic is to any of the conclusions.

In figure 6, plotting the results in D and E is non-intuitive. Instead of plotting the polar coordinates, a plot of the cartesian coordinates, corresponding to the strength of reward and delay encoding, would be more approachable.

Some of the theory terminology is inconsistent and it would help temper the conclusions if they were used consistently. Specifically, subjective value is first used to mean the behaviourally-calibrated model of value (e.g. on page 7). It is also used when describing signals that integrate reward and delay (e.g. on page 14). It is also equated to motivation in the discussion, which often refers to a different phenomenon, and if motivation is going to be discussed it should be defined explicitly.

Vector size and vector length can be confusing terms, as they may refer to the number of elements or the magnitude. Using magnitude throughout the text would resolve this.

Figure 1 H and I, the lower y-axis should either be labeled log(p) or have exponents of 10 marked on the axis but not both.

There is at least one missing reference from the bibliography (Yamada et al., 2021), cited in the methods section.

There are typos that need to be corrected, for example, page 22, line 8 "and a blank scream appeared".

*Reviewer #2 (Recommendations for the authors):*

There is good evidence that the subthalamic nucleus (STN) plays a role in self-control. The authors recorded from STN from monkeys that were trained to be still to obtain a reward. The STN seemed to integrate costs and benefits in this task to represent something like the utility of waiting.

The format of the paper – with captions separate from un-numbered figures added friction to the review process.

The data are "available" but there is no description of the data format, making the shared data not very useful.

I made the effort to understand the data_behavior.mat file. This only included per-session rejection rates, without detailed trial-by-trial information which might allow for deeper checks (order effects, reaction times, etc.). The other data are also shared at a very coarse level.

Overall, I found the claims were not strongly supported by the data. One flaw, that is impossible to correct with this dataset, is that the monkeys almost always wait. This means that the dataset does not permit a comparison of successful and unsuccessful trials of the same type. In addition, the analyses had other flaws (described in more detail in the public review and in the specific comments). I would advise the authors to read the series of papers by Murakami and Mainen (10.1038/nn.3826; j.neuron.2017.04.040) as examples of how to improve their visualizations and analyses.

Specific Comments:

– Abstract: "such that the most posterior-placed neurons represented the temporal discounting of value most strongly" this is not clear to me. I think "temporally discounted" is more accurate.

– The introduction could be improved. 'self-control' is overly broad. Waiting for a future reward is a very specific aspect of self-control (compared with other elements, e.g. the Barrett Impulsivity Scale) There is a massive literature on delay-discounting, and it is not all relevant. As the authors are aware, different tasks probe different dimensions of impulsivity. For example https://doi.org/10.1016/j.jaac.2009.12.018. I think the paper would be improved by a clearer statement of the specific aspects of "self-control" that the authors are addressing. For example, The ephys data was collected during a task that does not involve intertemporal choice. The subjects' only "choice" was to wait or not. Many studies of intertemporal choice "lock" the subjects in. Once the delayed option is chosen, the subject cannot change their mind. The neural mechanisms of this "locked-in" kind of intertemporal choice differ quite a bit from the mechanisms underlying "active waiting". The task described is even more strict than most – the subjects not only have to avoid making a specific action (e.g. pressing a button) but they have to suppress ALL action. This behavior seems to be quite far from the example they give – waiting in line at a bakery. While waiting in line, I can read a book, check email, etc. Interestingly, the authors did use a choice task during behavioral training – however, no data from this was shown.

– p3l18: If the delay is known, the subjective value of the future reward should grow as the subject gets ‘closer’ in time to the reward.

– Was the rejection rate predicted from the choice task behavior? i.e. what is the discount factor estimated by the choice task?

– Only analyzing rejection rates are insufficient. What is the distribution of wait times for each of the trial types? For example, if a monkey has a wait time distribution that only depended on reward size, then you would also see that the "rejection rate" was higher for the long delay.

– For the analysis in Figures 2 and 3, were only rewarded trials analyzed?

– In figure 2HJ and L the y-ticks are ‘fraction’ but the label says %, right?

– For figure 3C: this figure would be better if time was indicated by color or some other method.

– Including the eye "nuisance" variables could lead to spurious regression coefficients between task variables and STN activity if the task variables influence eye movement and STN influences eye movements. This is referred to as "collider bias" https://catalogofbias.org/biases/collider-bias/

– Although AIC is a common metric for model comparison, it is known to be flawed (10.1073/pnas.170283897). A better model comparison method would be the cross-validated likelihood (https://compneuro.neuromatch.io/tutorials/W1D2_ModelFitting/student/W1D2_Tutorial6.html?highlight=cross)

– It's a bit funny, from a narrative perspective, to say that your additive model (Eq 3) is better than the SV model (Eq 4), and then go on to say that overall the STN activity follows a pattern consistent with discounting.

– Were the neurons sampled from Anterior to Posterior over time? Could it be that the change that you have associated with the AP position is contaminated by changes in the animals' behavior over time?

– Figure 6C: same comment as above – need to represent time in some way. The colors for the pcs here don't really add anything, so you could use color for time.,

– I have concerns about the conclusion of increased delay encoding later in the waiting period. If neurons responded to the initiation of the waiting period with some long time-constant offset, then they would have time to return to baseline on long trials, but would not have time to return to baseline on short trials. This would appear (when analyzed relative to the reward) as a different level of activity on trials with different delays.

*Reviewer #3 (Recommendations for the authors):*

I have several comments on the statistic and analysis details. If the authors improve these issues, the manuscript must be more solid and pre-dominant.

Major comments (sequentially ordered as appeared in the manuscript)

1) P7 lines 16, corrected for 143-time bins.

I wonder why the author applied the correction of the type 1 error in this manuscript.

(1.1) The way of the multiple comparisons was not clearly written in the method section. Thus, no one can evaluate it.

(1.2) Is this too strong correction if the authors use Bonferroni-like correction?

(1.3) If they are interested in the dynamic change, it is unclear the reason why they need correction. If they focused on the moment-by-moment signals, the signals are not dependent on each other in their hypothesis testing. In this case, type 1 error correction underestimates the authors results.

The same issue exists for neural analyses.

2) P8, lines 3-5

The authors should show the fitting results of the temporal discounting function during decision makings (hence, choice?), which is the standard way to estimate the discount function. Please compare these discounted functions between two different tasks. It must be helpful.

3) P9, Lines 1-2

The authors corrected the type 1 error to show the percentage of modulated neurons by reward size and delay. This is the correct way to perform hypothesis testing to ask whether each single neuron encodes some information through a trial. However, the way of the correction was not clearly described in detail and it is unclear whether among 188 time-bins comparisons, the correction is too strong or not. I wonder whether these type 1 error corrections underestimated the neural modulations that occur in the different task periods. If I looked at Figure 3H, J, L. % neurons encoding delay or size looks below the 5% chance (especially for the size). If the authors focused on the dynamic changes of neural signal, this correction is inaccurate, since neural activity changed moment-by-moment, and thus, this is not the case to require multiple comparisons through a trial. I guess that without correction more signals would be detected, and those signals must important for arguing the dynamics at the population level in the conventional analyses, like changes of modulated neurons across times.

4) P10, line 15

Please present this result by describing figures, box plots plus each data plot with a line for paired data. I guess the mistake of the statistical test occurred since too small a difference between AIC values between models. In this case, a very strong tendency of the paired data is required to get this statistical significance level.

5) P10, lines 25-26

Please specify the underlying assumption for why the authors take summed vector across times (i.e., average). This is opposite to their story in that STN neurons dynamically change their coding of information. In dynamical coding, analysis of the average is meaningless. In this point of view, I recommend dividing the analysis window into half or four periods to take a vector sum in each period. If I looked at figure 3C, it would be reasonable. After dividing the analysis window, then the authors can perform the analysis in figure 4 for each of the analysis periods.

6) P12, Dynamic encoding part.

In this analysis using PCA, the authors need to use a non-overlapped analysis window, since PCA analyzes the variance and covariance matrix. If the overlapped analysis windows are used, similar activity modulations are enhanced more that the actual neural data because of the activity overlap in the analysis. This causes a serious problem in using PCA. Please use non-overlapped and discrete analysis windows, in which the neural dynamics are described appropriately.

7) Figure 5A and 7

Please discuss the finding related to somatotopic representation in the posterior part of the STN. It is clearly important to interpret the results with the anatomical connection to the prefrontal cortex (anterior STN) and somatotopic representation to the cortices (posterior STN).

[Editors' note: further revisions were suggested prior to acceptance, as described below.]

Thank you for resubmitting your work entitled "Neural dynamics underlying self-control in the primate subthalamic nucleus" for further consideration by *eLife*. Your revised article has been evaluated by Michael Frank (Senior Editor) and a Reviewing Editor.

The manuscript has been improved but there are some remaining issues that need to be addressed, as outlined below. We believe that these comments could be mostly addressed by adding some clarifications or by revising text in the manuscript.

*Reviewer 1:*

In general, I find the authors have addressed most of the points raised in the letter. I think of the major points I consider points 2 and 3 to be adequately resolved.

Regarding point 1, they have used the Bonferroni correction in their ANOVAs used in Figure 2. This is probably unnecessarily strong, but their results have not changed significantly. It is now at least adequately described and so I believe they have addressed this point in this part of their analysis. In determining the significance of the betas for the analysis that underpins the PCA approach, I am still not fully clear on what they did from their description of their process to determine the cutoff value. In their response letter, they have referenced it as a commonly used approach, so they could perhaps provide a reference that has a more detailed description. Ideally, they would provide the code for these steps.

As part of their response to point 3, the authors provided a supplement to Figure 4. The difference in the profile of the p-values is different between the monkeys and while this in itself may not be significant, it did make me notice that the authors do not report any per-monkey statistics beyond the behavior. While I do not fully understand the data that they have provided on GitHub, I had a brief look at it and I am concerned that there may be significant differences between the two animals, specifically that most of the significant neurons may be coming from one animal. This point was not raised previously in the review but I think having some indication of how consistent the findings were between the two animals is appropriate. The examples given in Figure 3 could be labelled which subject it came from.

*Reviewer 2:*

The authors did a reasonable job responding to the reviewers' comments. However, I find some of their claims still not well supported.

First, the abstract claims "This neural representation of subjective value evolved dynamically across the waiting period" – in the response to reviewers, the authors acknowledge that they do not have sufficient variability in task parameters to estimate the functional form of subjective value. Thus, the change in activity over time may not be a dynamic evolution of representation, but a stable representation of a dynamic subjective value. So, I think this sentence of the abstract should be removed. Second, Also the phrase "dynamically estimates" (in the penultimate sentence of the abstract )implies that the computation is in the dpSTN, but the authors write (in the discussion) that the STN might simply reflect the subjective value computed elsewhere. The last two sentences of the abstract could be edited to read "These findings highlight the selective role of the dpSTN in the representation of temporally discounted rewards. The integration of delays and rewards into an integrated representation is essential for self-control,.……"

If the abstract (and related content in the results/discussion) are updated as suggested (or similarly) such that they are consistent with the authors' updated manuscript, I could endorse the paper as a valuable contribution with solid evidence.

*Reviewer 3:*

The authors made significant improvements to their manuscript in general, and hence, their conclusion is now supported more strongly. However, I still have concerns about the two issues raised in the first revision.

1) Multiple comparison

This is a comment related to the Essential Revisions comment (1), and my comment (1) and (3) raised in the first revision.

After seeing the author's reply, I have further difficulty understanding whether their multiple comparison procedure is adequate or not. I will explain the detail below.

If I understand their procedure as their definition, Bonferroni correction methods give them p-value criterion at 0.05/n, where n is 175 (3.5/0.02) or at least more than 120. This means that the actual p-value used for detecting neural modulations is 0.00029 (0.05/175 or at least 0.0005) in each 200 ms time bin shifted every 20 ms. Then, they have these detections for all 175 (or more than 120) time bins that construct Figure 2G, I, and K. They said that in the methods "The threshold for significance in those ANOVAs was corrected for multiple comparisons using the Bonferroni correction (P < 0.05/ n‐time bins). The validity of that threshold was verified by calculating the likelihood of Type 1 (false‐positive) errors in a parallel analysis of activity extracted from a 1 s period of the pre‐instruction control epoch (i.e., from the time period when the animal had no information about the upcoming reward and delay)." This description may mean that type 1 error correction may not be performed exactly the same as Bonferroni correction, and they may use some other levels of type 1 error correction, though I cannot understand the detail. However, I guess that they may use just p/n correction as Bonferroni method, because they say "verified" above. This is one critical point where the author did not provide clear information in their manuscript.

In their reply, they said how they check whether this is appropriate correction by "The threshold for significance of individual β values was determined relative to a population of control coefficients calculated from a 1 s pre-instruction control epoch (one sample t-test, df=46, P < 0.05)." This is a critical issue because they used this correction for all of the neural and other response variable analyses, such as eye movement and EMG. The results depend on this correction, though I wonder why they find no significant neural modulation at the 1.0 sec pre-cue period (it would be found at around the 5% if they correctly improve the type 1 error) ? Is this too strong a correction, since n is more than 100?

If this point becomes clearer, I can understand the whole result, though I still do not evaluate this point well.

2) Dynamic analysis using overlapped analysis window.

This is the previous #6 comment of mine. In the first revision, they added the results for neural modulation dynamics with a discrete time window as in the supplemental figure for Figure 6. This is exactly the correct way to extract the neural dynamics to examine moment-by-moment signal change, and hence, this supplemental figure is better to be shown in the main text. Use of the overlapped analysis window is problematic theoretically and empirically if they further analyzed these dynamics. This is because overlapped extraction of information does not represent neural dynamics with fine time resolution. It is obvious that when analyzing the neural population dynamics, the smoothing in the dynamics analysis is totally different from the smoothing of information used in the conventional analysis the authors used in the first half of this manuscript. Thus, I strongly recommend presenting the results with a non-overlapped window in the main text.

However, the analysis with the non-overlapped time window may not give authors enough amount of population data for statistics (now it is 17 time points, 3.5 sec / 200ms). If they hope to present their results in more fine time resolution with non-overlapped time window, I recommend using the following procedure, without overlap.

1) First, please use spike density function for a bit wider distribution (like σ = 40 ms, now it is 20 ms) before applying regression analysis.

2) and then, applied a narrower time window without overlap (like 20 ms, not the 200 ms) which gives you 175 time points for (3.5 sec / 20ms).

If they use this procedure, then, they can extract finer dynamics without overlap.

The use of the overlapped time window is problematic. Please imagine the situation that the author met. They used 200 ms time window with 90 % overlap (20 ms shift). This means that if they define the time as 100 ms for the 0 to 200 ms bin (just take average time in the time bin). It does not give them precise dynamics at the moment. So, most of the study to extract neural population dynamics avoids using overlapped analysis window.

---

## [Author Response]

Essential revisions:The subthalamic nucleus (STN) is thought to play important roles in self-control. In this study, the authors sought to characterize the activity of STN neurons while monkeys performed a task in which they had to withhold their response during a delay period whose length was defined by a specific cue. The reviewers thought that this study contains potentially important results but they raised various concerns in some analyses, which reduced their enthusiasm for the present study. We hope that the authors can address the following essential issues.1) To identify neurons whose activity was modulated by reward size or delay, the authors analyzed many time bins (e.g. 188 sliding windows). The results must be properly corrected for multiple comparisons, yet the method used for correction is not fully described. It is therefore unclear whether the conclusions are well supported by the data.

We revised the text in the manuscript to clarify our methods. First, the threshold for significance in ANOVAs was corrected for multiple comparisons using the Bonferroni correction (P<0.05/n-time bins). The validity of that threshold (no overestimation as shown in Figure 2) was verified by calculating the likelihood of Type 1 errors during the control period preceding the instruction cues. We used that conservative approach to identify our task-related neurons before we tested how those selected neurons were influenced individually by task parameters over the course of the hold period. This second step in our analyses was performed with β coefficients calculated via multiple linear regressions (eq. 2). The threshold for significance of individual β values was determined relative to a population of control coefficients calculated from a 1 s pre-instruction control epoch (one sample *t*-test, df=46, *P* < 0.05). This statistical approach, commonly used in neurophysiological studies, is similar to methods used to detect changes in spike density functions, for instance. We agree with reviewer 3 – studying dynamic changes of β values as done here does not formally require a strong correction for multiple comparisons. In any case, independent of any statistical thresholds, we found consistent results using an alternative unsupervised approach (PCA), thereby suggesting that our single-unit analyses were appropriate for detecting task encodings.

2) The authors examined the STN activity aligned to the start of the delay and also aligned to the reward. Most of the "delay encoding" in the STN activity was observed near the end of the waiting period. The trouble with the analysis is that a neuron that responded with exactly the same response on short and long trials could appear to be modulated by delay. This is easiest to see with a diagram, but it should be easy to imagine a neural response that quickly rose at the time of instruction and then decayed slowly over the course of 2 seconds. For long trials, the neuron's activity would have returned to baseline, but for short trials, the activity would still be above baseline. As such, it is not clear how much the STN neurons were truly modulated by delay.

We agree with the reviewers. Our original analyses using two-time windows had the potential to introduce biases in the detection of neuronal activities modulated by the delay. To overcome this issue, we modified the time frame of all of our analyses (neuronal activity, eye position, EMG). Now, the revised version of the manuscript only reports activities across one-time window aligned to the time of instruction cue delivery (i.e., -1 to 3.5s relative to instruction cue onset). This time frame corresponds to the minimum possible interval between instruction cues and reward delivery. We have revised all of the figures and we re-calculated all of the statistics using that one analysis window. Despite these major modifications, our key findings were not changed substantially. We found the same pattern in STN activities, with a strong encoding of reward (48% of neurons) preceding a late encoding of delay (39% of neurons). We also updated the text in Methods and Results sections to reflect the revised analyses.

3) Another concern is the presence of eye movement variables in the regressions that determine whether a neuron is reward or delay encoding. If the task variables modulated eye movements (which would not be surprising) and if the STN activity also modulated eye movements, then, even if task variables did not directly modulate STN activity, the regression would indicate that it did. This is commonly known as "collider bias". This is, unfortunately, a common flaw in neuroscience papers.

Because the presence of eye variables did not influence how neurons were selected by the GLM, we do not think it likely that our analysis was susceptible to “collider bias”. Nonetheless, to control for that possibility directly, we have now repeated the GLM analyses with eye movement variables excluded. Results are shown in a new figure (Figure 4 – supplementary 1). Exclusion of eye parameters produced results that are very similar to those from the GLM that included eye parameters (differences <3 degrees). We have added text to the manuscript describing this added control analysis.

Reviewer #1 (Recommendations for the authors):There are a number of issues I noted:The claim "Accordingly, people with low discount rates tend to pursue their long-term goals patiently, whereas people with high discount rates often abandon their goals impulsively and move on." needs a citation.

We added a citation (pg. 3).

The phrasing of "Until now, however, existing evidence is mixed on whether the STN is causally involved in temporal discounting" implies that this paper may provide causal evidence, which it does not.

We revised the text (pg. 4)

The model of subjective value here assumes linearity in reward value; this should be made explicit in the text and supported by any evidence you have. It also assumes that the asymptotic rejection rate is zero, which may bias the estimate of k.

We revised the text in the Methods section (pg. 26)

No analysis relates to the transition from the gray-filled circle to the unfilled circle and it is unclear of the relevance of this visual cue to the task design.It is unclear what 'as a control' means when comparing the fits of the models described in Eq. 3 and Eq. 4 or what the relevance of this statistic is to any of the conclusions.

We removed this part of the analyses because it did not add to the conclusions of the paper.

Some of the theory terminology is inconsistent and it would help temper the conclusions if they were used consistently. Specifically, subjective value is first used to mean the behaviourally-calibrated model of value (e.g. on page 7). It is also used when describing signals that integrate reward and delay (e.g. on page 14). It is also equated to motivation in the discussion, which often refers to a different phenomenon, and if motivation is going to be discussed it should be defined explicitly.

We revised the text make the use of terms more consistent.

Vector size and vector length can be confusing terms, as they may refer to the number of elements or the magnitude. Using magnitude throughout the text would resolve this.

We revised the text following this advice.

Figure 1 H and I, the lower y-axis should either be labeled log(p) or have exponents of 10 marked on the axis but not both.

We changed the labels in Figure 1.

There is at least one missing reference from the bibliography (Yamada et al., 2021), cited in the methods section.

We added the reference.

There are typos that need to be corrected, for example, page 22, line 8 "and a blank scream appeared".

We revised the text.

Reviewer #2 (Recommendations for the authors):There is good evidence that the subthalamic nucleus (STN) plays a role in self-control. The authors recorded from STN from monkeys that were trained to be still to obtain a reward. The STN seemed to integrate costs and benefits in this task to represent something like the utility of waiting.The format of the paper – with captions separate from un-numbered figures added friction to the review process.The data are "available" but there is no description of the data format, making the shared data not very useful.I made the effort to understand the data_behavior.mat file. This only included per-session rejection rates, without detailed trial-by-trial information which might allow for deeper checks (order effects, reaction times, etc.). The other data are also shared at a very coarse level.Overall, I found the claims were not strongly supported by the data. One flaw, that is impossible to correct with this dataset, is that the monkeys almost always wait. This means that the dataset does not permit a comparison of successful and unsuccessful trials of the same type. In addition, the analyses had other flaws (described in more detail in the public review and in the specific comments). I would advise the authors to read the series of papers by Murakami and Mainen (10.1038/nn.3826; j.neuron.2017.04.040) as examples of how to improve their visualizations and analyses.

Our main goal was not to investigate action timing in the STN. This question could be addressed in future research.

Specific Comments:– Abstract: "such that the most posterior-placed neurons represented the temporal discounting of value most strongly" this is not clear to me. I think "temporally discounted" is more accurate.

We revised the text.

– The introduction could be improved. 'self-control' is overly broad. Waiting for a future reward is a very specific aspect of self-control (compared with other elements, e.g. the Barrett Impulsivity Scale) There is a massive literature on delay-discounting, and it is not all relevant. As the authors are aware, different tasks probe different dimensions of impulsivity. For example https://doi.org/10.1016/j.jaac.2009.12.018. I think the paper would be improved by a clearer statement of the specific aspects of "self-control" that the authors are addressing. For example, The ephys data was collected during a task that does not involve intertemporal choice. The subjects' only "choice" was to wait or not. Many studies of intertemporal choice "lock" the subjects in. Once the delayed option is chosen, the subject cannot change their mind. The neural mechanisms of this "locked-in" kind of intertemporal choice differ quite a bit from the mechanisms underlying "active waiting". The task described is even more strict than most – the subjects not only have to avoid making a specific action (e.g. pressing a button) but they have to suppress ALL action. This behavior seems to be quite far from the example they give – waiting in line at a bakery. While waiting in line, I can read a book, check email, etc.

Many studies have addressed the psychology of queuing in situations where alternative activities were not allowed. We believe that long wait times impose a cost that requires patience to overcome, even if alternative activities are allowed. In addition, if someone cannot wait more than 5 minutes in a queue without engaging in alternative activities (benefits), then that means that his patience is limited.

Interestingly, the authors did use a choice task during behavioral training – however, no data from this was shown.

In the Results section, there is a paragraph describing results obtained from this choice task (pgs. 7-8).

– p3l18: If the delay is known, the subjective value of the future reward should grow as the subject gets ‘closer’ in time to the reward.

We revised the text (pg. 3)

– Was the rejection rate predicted from the choice task behavior? i.e. what is the discount factor estimated by the choice task?

The choice task did not induce similar rejection rates. During this choice task, monkeys choose freely between two reward size or delay conditions. Decisions were made only in one dimension at a time (between 2 rewards or between 2 delays). In other words, the choice task did not require animals to integrate the two task parameters to make decisions. Hence, no discounting function could be calculated based on that task.

– Only analyzing rejection rates are insufficient. What is the distribution of wait times for each of the trial types? For example, if a monkey has a wait time distribution that only depended on reward size, then you would also see that the "rejection rate" was higher for the long delay.– For the analysis in Figures 2 and 3, were only rewarded trials analyzed?

As indicated in the Methods section (pg. 26), trials with errors were excluded from the analysis of neuronal data.

– In figure 2HJ and L the y-ticks are ‘fraction’ but the label says %, right?

We modified the y-axis in Figure 2.

– For figure 3C: this figure would be better if time was indicated by color or some other method.

We modified the figure 3C with yellow-to-black lines to indicate the passage of time.

– Including the eye "nuisance" variables could lead to spurious regression coefficients between task variables and STN activity if the task variables influence eye movement and STN influences eye movements. This is referred to as "collider bias" https://catalogofbias.org/biases/collider-bias/

See response above. We revised the text and Figure 4 – supplementary 1 to address this point.

– Although AIC is a common metric for model comparison, it is known to be flawed (10.1073/pnas.170283897). A better model comparison method would be the cross-validated likelihood

We removed this part of the analysis because it did not add anything to the conclusions of the paper. In addition, all of the methods used to compare models have their own flaws.

(https://compneuro.neuromatch.io/tutorials/W1D2_ModelFitting/student/W1D2_Tutorial6.html?highlight=cross)– It's a bit funny, from a narrative perspective, to say that your additive model (Eq 3) is better than the SV model (Eq 4), and then go on to say that overall the STN activity follows a pattern consistent with discounting.– Were the neurons sampled from Anterior to Posterior over time? Could it be that the change that you have associated with the AP position is contaminated by changes in the animals' behavior over time?

Electrode trajectories through the STN were not changed in a systematic fashion between sessions or monkeys. Furthermore, we find no correlation between the date of recording and the type of task-related single-unit activity found.

– Figure 6C: same comment as above – need to represent time in some way. The colors for the pcs here don't really add anything, so you could use color for time.,

We modified the figure 6 with gradient colors to indicate the time.

– I have concerns about the conclusion of increased delay encoding later in the waiting period. If neurons responded to the initiation of the waiting period with some long time-constant offset, then they would have time to return to baseline on long trials, but would not have time to return to baseline on short trials. This would appear (when analyzed relative to the reward) as a different level of activity on trials with different delays.Reviewer #3 (Recommendations for the authors):I have several comments on the statistic and analysis details. If the authors improve these issues, the manuscript must be more solid and pre-dominant.Major comments (sequentially ordered as appeared in the manuscript)1) P7 lines 16, corrected for 143-time bins.I wonder why the author applied the correction of the type 1 error in this manuscript.(1.1) The way of the multiple comparisons was not clearly written in the method section. Thus, no one can evaluate it.(1.2) Is this too strong correction if the authors use Bonferroni-like correction?(1.3) If they are interested in the dynamic change, it is unclear the reason why they need correction. If they focused on the moment-by-moment signals, the signals are not dependent on each other in their hypothesis testing. In this case, type 1 error correction underestimates the authors results.The same issue exists for neural analyses.

See the answer above to comment #1 in Essential Revisions.

2) P8, lines 3-5The authors should show the fitting results of the temporal discounting function during decision makings (hence, choice?), which is the standard way to estimate the discount function. Please compare these discounted functions between two different tasks. It must be helpful.

We revised the text to be more clear. Animals were trained to perform a decision task in which they were allowed to choose between two alternate reward size or delay conditions. Decisions were made only along one dimension of the task at a time (between 2 rewards or between 2 delays). Because of that, the decision-making (choice) task did not require animals to integrate the two task parameters to make decisions. Hence, no discounting function could be calculated based on the decision-making task.

3) P9, Lines 1-2The authors corrected the type 1 error to show the percentage of modulated neurons by reward size and delay. This is the correct way to perform hypothesis testing to ask whether each single neuron encodes some information through a trial. However, the way of the correction was not clearly described in detail and it is unclear whether among 188 time-bins comparisons, the correction is too strong or not. I wonder whether these type 1 error corrections underestimated the neural modulations that occur in the different task periods. If I looked at Figure 3H, J, L. % neurons encoding delay or size looks below the 5% chance (especially for the size). If the authors focused on the dynamic changes of neural signal, this correction is inaccurate, since neural activity changed moment-by-moment, and thus, this is not the case to require multiple comparisons through a trial. I guess that without correction more signals would be detected, and those signals must important for arguing the dynamics at the population level in the conventional analyses, like changes of modulated neurons across times.

See response to comment #1 in Essential Revisions. The selection of an adequate statistical threshold can be tricky. Reviewer #1 thinks that our method risked inducing an overestimation of the number of task-dependent neurons, while reviewer #3 suggests an underestimation.

4) P10, line 15Please present this result by describing figures, box plots plus each data plot with a line for paired data. I guess the mistake of the statistical test occurred since too small a difference between AIC values between models. In this case, a very strong tendency of the paired data is required to get this statistical significance level.

We removed the AIC part of the analyses because it did not strengthen the conclusions of the paper. In addition, the method used was criticized for different reasons by different reviewers.

5) P10, lines 25-26Please specify the underlying assumption for why the authors take summed vector across times (i.e., average). This is opposite to their story in that STN neurons dynamically change their coding of information. In dynamical coding, analysis of the average is meaningless. In this point of view, I recommend dividing the analysis window into half or four periods to take a vector sum in each period. If I looked at figure 3C, it would be reasonable. After dividing the analysis window, then the authors can perform the analysis in figure 4 for each of the analysis periods.

We revised the manuscript and figures 3-5 by splitting the waiting period into two intervals (phase 1 and phase 2). This new approach allowed us to describe key aspects of the dynamic changes in coding of task information as a function of elapsed time during the waiting period. These results, based on single-unit analyses, appear quite consistent with the results obtained from the PCA. We added paragraphs in the Results section to explain this new approach in detail.

6) P12, Dynamic encoding part.In this analysis using PCA, the authors need to use a non-overlapped analysis window, since PCA analyzes the variance and covariance matrix. If the overlapped analysis windows are used, similar activity modulations are enhanced more that the actual neural data because of the activity overlap in the analysis. This causes a serious problem in using PCA. Please use non-overlapped and discrete analysis windows, in which the neural dynamics are described appropriately.

To address this point, we added a new figure and we revised the text in the Methods section (pg. 30). We compared our overlapping window PCA results against those from a PCA using non-overlapped temporal windows in the estimation of regression coefficients. As shown on the Figure 6 – supplement 1, results were very similar to those from the sliding window analysis (see % of variance explained per PC). Consequently, our results were not biased by the method used.

7) Figure 5A and 7Please discuss the finding related to somatotopic representation in the posterior part of the STN. It is clearly important to interpret the results with the anatomical connection to the prefrontal cortex (anterior STN) and somatotopic representation to the cortices (posterior STN).

Unfortunately, our approach and the literature does not allow us to be more specific about the specificity of different STN territories. Despite the potential overlaps between territories, the tripartite subdivision is the most popular model of STN functional topography. In the Discussion section, our paragraph (pgs. 19-20) about the anatomo-functional organization of the STN interprets our findings based on anatomo-functional considerations.

[Editors' note: further revisions were suggested prior to acceptance, as described below.]

Thank you for resubmitting your work entitled "Neural dynamics underlying self-control in the primate subthalamic nucleus" for further consideration by eLife. Your revised article has been evaluated by Michael Frank (Senior Editor) and a Reviewing Editor.The manuscript has been improved but there are some remaining issues that need to be addressed, as outlined below. We believe that these comments could be mostly addressed by adding some clarifications or by revising text in the manuscript.Reviewer 1:In general, I find the authors have addressed most of the points raised in the letter. I think of the major points I consider points 2 and 3 to be adequately resolved.Regarding point 1, they have used the Bonferroni correction in their ANOVAs used in Figure 2. This is probably unnecessarily strong, but their results have not changed significantly. It is now at least adequately described and so I believe they have addressed this point in this part of their analysis. In determining the significance of the betas for the analysis that underpins the PCA approach, I am still not fully clear on what they did from their description of their process to determine the cutoff value. In their response letter, they have referenced it as a commonly used approach, so they could perhaps provide a reference that has a more detailed description. Ideally, they would provide the code for these steps.

We revised the text of the Methods section to clarify this point.

As part of their response to point 3, the authors provided a supplement to Figure 4. The difference in the profile of the p-values is different between the monkeys and while this in itself may not be significant, it did make me notice that the authors do not report any per-monkey statistics beyond the behavior. While I do not fully understand the data that they have provided on GitHub, I had a brief look at it and I am concerned that there may be significant differences between the two animals, specifically that most of the significant neurons may be coming from one animal. This point was not raised previously in the review but I think having some indication of how consistent the findings were between the two animals is appropriate. The examples given in Figure 3 could be labelled which subject it came from.

In Results section (pg.9), we revised the text to include results for each monkey. Despite an asymmetry between animals, likely reflecting differences in the density of sampling from the dorso-posterior STN, we found neurons with the main types of task-encoding in both monkeys.

Reviewer 2:The authors did a reasonable job responding to the reviewers' comments. However, I find some of their claims still not well supported.First, the abstract claims "This neural representation of subjective value evolved dynamically across the waiting period" – in the response to reviewers, the authors acknowledge that they do not have sufficient variability in task parameters to estimate the functional form of subjective value. Thus, the change in activity over time may not be a dynamic evolution of representation, but a stable representation of a dynamic subjective value. So, I think this sentence of the abstract should be removed. Second, Also the phrase "dynamically estimates" (in the penultimate sentence of the abstract )implies that the computation is in the dpSTN, but the authors write (in the discussion) that the STN might simply reflect the subjective value computed elsewhere.

We agree. Our study was not designed to identify the source of the dynamic evolution observed. We have revised the text of the manuscript to express a more agnostic view. More specifically, previous references to a “dynamic evolution of representation” have been reworded.

The last two sentences of the abstract could be edited to read "These findings highlight the selective role of the dpSTN in the representation of temporally discounted rewards. The integration of delays and rewards into an integrated representation is essential for self-control,.……"

We have revised the Abstract as suggested.

Reviewer 3:The authors made significant improvements to their manuscript in general, and hence, their conclusion is now supported more strongly. However, I still have concerns about the two issues raised in the first revision.1) Multiple comparisonThis is a comment related to the Essential Revisions comment (1), and my comment (1) and (3) raised in the first revision.After seeing the author's reply, I have further difficulty understanding whether their multiple comparison procedure is adequate or not. I will explain the detail below.If I understand their procedure as their definition, Bonferroni correction methods give them p-value criterion at 0.05/n, where n is 175 (3.5/0.02) or at least more than 120. This means that the actual p-value used for detecting neural modulations is 0.00029 (0.05/175 or at least 0.0005) in each 200 ms time bin shifted every 20 ms. Then, they have these detections for all 175 (or more than 120) time bins that construct Figure 2G, I, and K. They said that in the methods "The threshold for significance in those ANOVAs was corrected for multiple comparisons using the Bonferroni correction (P < 0.05/ n‐time bins). The validity of that threshold was verified by calculating the likelihood of Type 1 (false‐positive) errors in a parallel analysis of activity extracted from a 1 s period of the pre‐instruction control epoch (i.e., from the time period when the animal had no information about the upcoming reward and delay)." This description may mean that type 1 error correction may not be performed exactly the same as Bonferroni correction, and they may use some other levels of type 1 error correction, though I cannot understand the detail. However, I guess that they may use just p/n correction as Bonferroni method, because they say "verified" above. This is one critical point where the author did not provide clear information in their manuscript.

Reviewer 3 is correct to surmise that we used the simple p/n Bonferroni correction (and no extra correction). We have revised the cited section of Methods to read “Application of the same ANOVA analysis and statistical threshold (p<0.05/n-time bins) to neuronal data during the 1 s period of the pre-instruction control epoch (i.e., from a time period when the animal had no information about the upcoming reward and delay) resulted in 0.12% of Type 1 (false-positive) errors, thereby confirming that the threshold for statistical significance was appropriate.”

In their reply, they said how they check whether this is appropriate correction by "The threshold for significance of individual β values was determined relative to a population of control coefficients calculated from a 1 s pre-instruction control epoch (one sample t-test, df=46, P < 0.05)." This is a critical issue because they used this correction for all of the neural and other response variable analyses, such as eye movement and EMG. The results depend on this correction, though I wonder why they find no significant neural modulation at the 1.0 sec pre-cue period (it would be found at around the 5% if they correctly improve the type 1 error) ? Is this too strong a correction, since n is more than 100?

We have revised the manuscript to clarify that the (p<0.05/n-time bins) correction was used only for the ANOVA analyses. A different method was used to determine the significance of results from the linear regression analyses. More specifically, the significance of individual test β values was determined by comparing them against a population of 46 “control” β values calculated from a 1 second-long control epoch prior to the onset of the instruction cue. That approach precluded testing for Type 1 errors during the 1.0 sec pre-cue period.

2) Dynamic analysis using overlapped analysis window.This is the previous #6 comment of mine. In the first revision, they added the results for neural modulation dynamics with a discrete time window as in the supplemental figure for Figure 6. This is exactly the correct way to extract the neural dynamics to examine moment-by-moment signal change, and hence, this supplemental figure is better to be shown in the main text. Use of the overlapped analysis window is problematic theoretically and empirically if they further analyzed these dynamics. This is because overlapped extraction of information does not represent neural dynamics with fine time resolution. It is obvious that when analyzing the neural population dynamics, the smoothing in the dynamics analysis is totally different from the smoothing of information used in the conventional analysis the authors used in the first half of this manuscript. Thus, I strongly recommend presenting the results with a non-overlapped window in the main text.However, the analysis with the non-overlapped time window may not give authors enough amount of population data for statistics (now it is 17 time points, 3.5 sec / 200ms). If they hope to present their results in more fine time resolution with non-overlapped time window, I recommend using the following procedure, without overlap.1) First, please use spike density function for a bit wider distribution (like σ = 40 ms, now it is 20 ms) before applying regression analysis.2) and then, applied a narrower time window without overlap (like 20 ms, not the 200 ms) which gives you 175 time points for (3.5 sec / 20ms).If they use this procedure, then, they can extract finer dynamics without overlap.The use of the overlapped time window is problematic. Please imagine the situation that the author met. They used 200 ms time window with 90 % overlap (20 ms shift). This means that if they define the time as 100 ms for the 0 to 200 ms bin (just take average time in the time bin). It does not give them precise dynamics at the moment. So, most of the study to extract neural population dynamics avoids using overlapped analysis window.

We have run a new analysis using non-overlapping and shorter during time windows. In the revised manuscript, results from that analysis are shown in new Figures 6 and 7. It is worth noting that the patterns of evolving population encoding revealed in the original Figure 6, which was produced using overlapping analysis windows, are preserved in nearly every detail in this new analysis. The new analysis did reveal a significant anatomic gradient for principal component 2 as described in revised text and illustrated in the revised Figure 7.